# Deep topographic proteomics of a human brain tumour

Simon Davis [1,2], Connor Scott [3], Janina Oetjen [4], Philip D. Charles [1,5], Benedikt M. Kessler [1,2], Olaf Ansorge[3] & Roman Fischer [1,2] ✉

The spatial organisation of cellular protein expression profiles within tissue determines cellular function and is key to understanding disease pathology. To define molecular phenotypes in the spatial context of tissue, there is a need for unbiased, quantitative technology capable of mapping proteomes within tissue structures. Here, we present a workflow for spatially-resolved, quantitative proteomics of tissue that generates maps of protein abundance across tissue slices derived from a human atypical teratoid-rhabdoid tumour at three spatial resolutions, the highest being 40 μm, to reveal distinct abundance patterns of thousands of proteins. We employ spatially-aware algorithms that do not require prior knowledge of the fine tissue structure to detect proteins and pathways with spatial abundance patterns and correlate proteins in the context of tissue heterogeneity and cellular features such as extracellular matrix or proximity to blood vessels. We identify PYGL, ASPH and CD45 as spatial markers for tumour boundary and reveal immune response-driven, spatially-organised protein networks of the extracellular tumour matrix. Overall, we demonstrate spatially-aware deep proteo-phenotyping of tissue heterogeneity, to re-define understanding tissue biology and pathology at the molecular level.

Tissues are composed of various microscopic features, cell types, and phenotypically diverse subpopulations, the location of the cells within a tissue and their spatial neighbourhood is crucial for determining their identity and function[1–6]. The cellular composition of tissue has a substantial influence on measured co-expression signals within the molecular profiles of bulk tissue and their micro-environment, contributing to the influence of cellular function, signalling and different disease outcomes[7–11]. Recent technological developments in spatially-resolved sequencing technologies have enabled the characterisation of spatially heterogeneous gene expression profiles within tissues[12,13]. However, while genomic and transcriptomic alterations act as drivers of disease, the proteins they encode regulate essentially all cellular processes and therefore

also need consideration when investigating tissue spatial heterogeneity[14].

A range of mass spectrometry (MS)-based techniques are available to map the distribution of proteins throughout tissues and cells. Mass spectrometry imaging (MSI) enables the determination of proteins or other molecules within a sample by rastering an ion source over a sample in a grid pattern. MSI techniques to visualise biomolecules in situ cannot generate in-depth proteome data and may require prior knowledge of measurement targets[15–18].

Laser capture microdissection (LCM) is well-placed to address the limitations of the spatially-resolved mass spectrometry methods described above[19]. LCM allows the extraction of regions from a tissue slice ranging from single cells to square millimetres of tissue[20,21].

[1]Target Discovery Institute, Centre for Medicines Discovery, Nuffield Department of Medicine, University of Oxford, Roosevelt Drive, Oxford OX3 7FZ, UK. [2]Chinese Academy for Medical Sciences Oxford Institute, Nuffield Department of Medicine, University of Oxford, Roosevelt Drive, Oxford OX3 7FZ, UK. [3]Academic Unit of Neuropathology, Nuffield Department of Clinical Neurosciences, University of Oxford, John Radcliffe Hospital, Oxford OX3 9DU, UK. [4]Bruker Daltonics GmbH & Co. KG, Fahrenheitstraße 4, 28359 Bremen, Germany. [5]Big Data Institute, Nuffield Department of Medicine, University of Oxford, Roosevelt Drive, Oxford OX3 7FZ, UK. ✉e-mail: roman.fischer@ndm.ox.ac.uk

We and others have previously described several methods that either couple LCM to MS-based proteomics[22–26], use micro-scaffolds[27] or tissue expansion followed by punch biopsies[28]. These sampling approaches have been used to investigate a wide range of tissue biology, but typically follow a 'feature-driven' approach, extracting tissue regions based on visible features[29–32]. Recently, Mund et al. developed the concept of Deep Visual Proteomics, which combines high-resolution imaging and machine learning-based image analysis to classify, isolate and analyse cells using a sensitive proteomics workflow[33].

However, sampling in a systematic manner, like MSI, could reveal novel tissue fine-structure and give insights into spatial protein expression patterns. For instance, Petyuk et al. sampled seventy, 1 mm cubes of mouse brain to generate a spatial proteome at a depth of approximately 1000 proteins[34]. Piehowski et al. used LCM-proteomics to sample a feature-rich landscape of mouse uterine tissue in a rastered grid at a resolution of 100 μm and used their custom, robotic, nanolitre-scale nanoPOTS sample preparation platform to quantify over 2000 proteins across 24 voxels[35]. Additionally, Ma et al. used a micro-scaffold to cut 1 mm-thick sections of mouse brain at 400 μm resolution prior to quantifying 5000 proteins using LC-MS/MS[27]. However, these approaches have primarily emphasised visualising protein abundance, neglecting the potential of utilising spatial relationships between areas of correlated protein expression and have been unable to discover novel spatial features, an approach frequently employed in spatial transcriptomics methodologies.

In this study, we conduct a systematic analysis of the proteome of a human brain tumour (Supplementary Fig. 1), using laser capture microdissection on three length scales, ranging from coverage of an entire tumour tissue section, down to 40 μm spatial resolution (approximately 10 cells/voxel), allowing for the proteomic mapping of the tumour microenvironment by spatially-resolved measurements. The principal aim of the study is to design a workflow for deep proteomics and spatially-aware statistics for tissue feature discovery, rather than provide an in depth analysis of atypical teratoid/rhabdoid tumour (AT/RT) biology, which will be the subject of a follow-up 'multiomics' study on paediatric brain tumours. The AT/RT is chosen based on tissue quality, homogeneity and abundance. By applying spatially-aware statistical methods, we identify proteins and pathways with differential spatial and clustered expression within the tissue sections, without prior knowledge of tissue structures, features, or pathology. Additionally, by clustering protein expression, we discover spatially-defined proteo-phenotypes within the otherwise homogeneous macrostructure of the analysed tumour, revealing spatially-resolved extracellular matrix (ECM) biology correlated with a focussed immune response spatially limited to the periphery of the tumour. Further, we map protein abundances in spatial dependence of the vascular fine structure in the tumour, giving a glimpse on the nutrient/oxygen dependent spatial proteome within cancerous tissue. Our work demonstrates that deep topographic proteomics can be used beyond confirming the differential proteome in observable features and - when combined with spatially-aware analysis - identify areas of potential interest regarding disease mechanisms.

## Results

In order to establish the required tissue area size to achieve a target proteome depth of 4000 quantified proteins in low (Orbitrap Fusion Lumos, 60-minute gradient) and medium throughput proteomics platforms (TimsTOF Pro, 17-minute gradient) we collected tissue areas ranging from 316 μm² to 1,000,000 μm² from 10 μm-thick sections of human brain (Supplementary Fig. 2). We observed that areas above 316,000 μm² result in diminishing returns with protein identifications scaling between 282 and 3480 on the Orbitrap and 127 and 3318 on the timsTOF Pro platforms, respectively.

### Proteomic topography of a human brain tumour

After characterising the upper and lower limits of our workflow (Fig. 1), we sampled a 10 μm thick section of an atypical teratoid-rhabdoid tumour (AT/RT) block (~20 × 15 mm), retrieved at postmortem and frozen in liquid nitrogen vapour. The tumour was fully characterised in vivo, corresponding to a supratentorial AT/RT with nuclear loss of SMARCB1 protein and corresponding to methylation class AT/RT-SHH (Supplementary Fig. 1). Balancing proteome depth with throughput and feasibility, we decided to approach deep topographic proteomics first with a coarse resolution to cover the total tumour and peripheral tissue. The tissue was subdivided into 384 (24 × 16) square 'voxels' with an area of ~694,000 μm² (side length of 833 μm and thickness of 10 μm); each voxel was isolated by LCM and processed with our LCM-SP3 protocol[24] (Fig. 1, Supplementary Data 1) and analysed on the medium throughput timsTOF Pro setup. In total, 5321 proteins were identified, with 32–4741 proteins identified per sample (Supplementary Fig. 3a). This range of proteins identified per sample includes empty voxels where no tissue was located within a voxel, demonstrating a low level of contamination throughout the workflow. These empty voxels are included in the dataset identification overview in Supplementary Fig. 3b.

Quantified proteins can be mapped back to their original positions within the tissue grid. Figure 2a shows proteomic maps for four example proteins, haemoglobin (HBB), histone H4 (HIST1H4A), peripherin (PRPH) and liver glycogen phosphorylase (PYGL). The selected proteins were chosen as examples showing: positive autocorrelation and increased abundance in tumour (PYGL) or periphery (PRPH); no autocorrelation (HIST1H4A, indicating similar DNA abundance/cell numbers); and a protein which should correlate with the large region of haemorrhage visible in the upper-right region of the section (HBB). Glycogen phosphorylase releases glucose from glycogen for entry into glycolysis, and its expression in cancer is associated with malignant phenotypes, hypoxia resistance and cancer cell survival[36]. Peripherin is an intermediate filament protein without a clear function and is highly expressed during development and after nerve injury; its expression pattern is consistent with the tumour growth into surrounding normal brain tissue[37–39].

We tested 4306 proteins quantified in at least 9 voxels for spatial autocorrelation[40] using the unsupervised Moran's $I$ test. Moran's I returns values between −1 for complete dispersion and +1 for complete correlation, with 0 indicating random distribution. Removing empty voxels and areas of haemorrhage from the analysis, we found that 3212 proteins demonstrated correlated spatial expression profiles, ($q \le 0.05$) indicating a low level of spatial noise in protein distribution across a tissue. To determine whether the observed spatial variability could have been caused by systematic impacts derived from tissue sampling and processing, the summed and mean intensity per voxel were inspected and are shown to be visibly consistent across the tissue section (Supplementary Fig. 4a, b). This demonstrates that no sampling bias was introduced and protein abundance differences are mostly compositional.

To increase spatial resolution, particularly in the 'brain/tumour' interface as identified by H&E staining, we sampled 96 voxels with 350 μm x 350 μm voxel side-length and 10 μm thickness (Fig. 2b, Supplementary Data 2). This region of tissue contains predominantly normal and neoplastic cells along with a large, prominent blood vessel. In total, 3994 proteins were quantified in at least one voxel. Detailed identification and quantification information is shown in Supplementary Fig. 3c for proteins identified and quantified per voxel, and in Supplementary Fig. 3d for the distributions of how many voxels each protein was identified and quantified in. The increased resolution proteomic maps for haemoglobin, histone H4, PRPH and PYGL are shown in Fig. 2b. Their expression is consistent with the large field-of-view data, with PYGL and PRPH showing opposite expression patterns

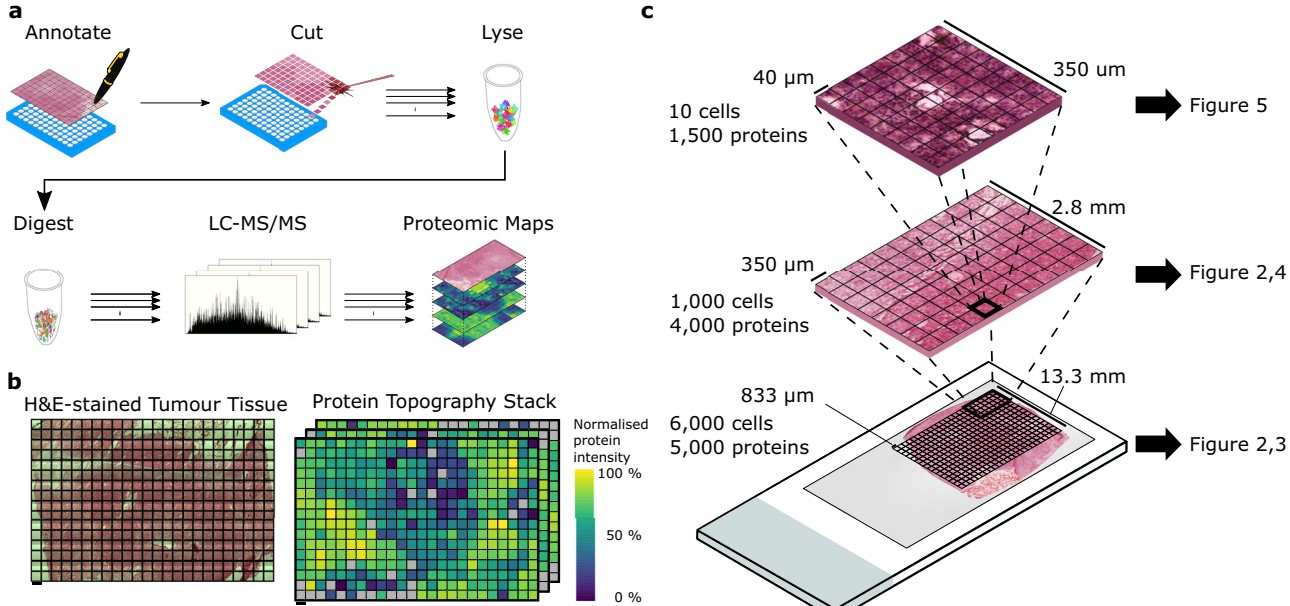

**Fig. 1 | Overview of the spatially-resolved proteomics workflow.** Tissue is mounted onto a slide compatible with laser capture microdissection (LCM). A general overview is depicted in panel **a**. Tissue is segmented into a regular grid shape (Annotate), and each element of the grid is isolated by LCM into a well of a 96-well plate (Cut). Proteins from each sample are lysed in RIPA buffer (Lyse) and digested into peptides (Digest) before analysis by LC-MS/MS. The quantitative information for each protein can be mapped back to its location within the gridded tissue and visualised in a topographic protein map, with one map per protein quantified (Proteomic Maps). This workflow was applied to an Atypical Teratoid-Rhabdoid Tumour (AT/RT). **b** A H&E-stained section was segmented into a 24×16 grid and analysed with the workflow to generate a Protein Topography Stack

containing over 5,000 proteomic maps at 833 μm resolution, allowing for the resolution of several features within the tissue while maintaining good throughput. **c** We then proceeded to apply this workflow at smaller length scales. In total, we applied this workflow over three length scales within serial sections of the AT/RT tumour tissue: 833 μm resolution, covering an entire tumour section; 350 μm resolution, covering part of the boundary between two visibly distinct regions; and 40 μm resolution, covering several blood vessels and their surrounding cells. These data relating to these three length scales are shown in Figs. 2, 3, 4 and 5, respectively. Scale bar represents 833 μm in the bottom, 350 μm in the middle image, and 40 μm in the top image.

across the margin between solid tumour (high PYGL, low PRPH) and brain/tumour interface (low PYGL, high PRPH) and haemoglobin co-localising with the visible blood vessels. Histone H4 shows even expression across the two annotated areas, with a region of lower expression corresponding with a visibly diffuse patch of tissue. Of the 3050 proteins quantified in at least 9 voxels, 1375 show evidence for significant spatial autocorrelation (Moran's *I* test, $q \leq 0.05$).

Three proteins showing significant spatial variation were selected for follow-up immunohistochemistry (IHC) staining and are presented side-by-side with their proteomic maps: glycogen phosphorylase (Supplementary Fig. 5a, b), aspartate beta-hydroxylase (ASPH) (Supplementary Fig. 5c, d) and CD45 (PTPRC) (Supplementary Fig. 5e, f) to validate the spatially-resolved protein expression data generated above. The location within the proteomic maps of the presented IHC images are marked by the black boxes. The IHC staining images closely resemble the protein intensity distributions (Supplementary Fig. 5a, c, e) within the proteomic maps for these three proteins. Both PYGL and ASPH show intense IHC staining in the solid tumour (Supplementary Fig. 5b, d), and CD45 shows intense staining in the region of tissue corresponding to the upper-left voxels in the proteomic map (Supplementary Fig. 5f).

### Spatial proteomic mapping highlights molecular pathways underlying tissue heterogeneity

As Moran's *I* measures global autocorrelation, it does not indicate where the locations that drive the autocorrelation occur. To investigate which regions of the sampled tissue show similar expression, the data were clustered and the cluster labelling of voxels were mapped back to their spatial location (Fig. 3a, b). The clusters generally form contiguous regions in space, with some long-range co-clustering in smaller clusters.

The margin between solid tumour and brain/tumour interface is well-represented by the border between cluster 1 (solid tumour) and cluster 3 (brain/tumour interface). In addition to the cluster map, the assigned clusters were plotted onto a uniform manifold approximation and projection (UMAP) visualisation (Fig. 3c). The clusters visible in the UMAP plot correspond well to the spatial clusters in Fig. 3b.

This clustering approach generates spatially well-defined distinct areas, allowing for a feature-driven approach without prior knowledge of the histopathological details as the clustering is performed only on the protein quantitation data, without information on the spatial relationship between samples. The functional differences between these clusters were investigated by first determining 'marker proteins' for each cluster by testing for differential abundance of a protein in one cluster versus all other clusters, iteratively for each protein-cluster pair. Protein markers for each cluster were then tested for functional enrichment against the MSigDB Hallmark gene sets (Fig. 3d)[41]. Clusters 1 & 2, the largest clusters within the solid tumour, show enrichment for proliferative hallmarks. Clusters 6 & 13, two neighbouring clusters in the upper-left region, show enrichment for immune-related hallmarks. Cluster 9, a cluster on the boundary of the tumour, shows enrichment of hallmarks related to angiogenesis and epithelial-mesenchymal transition, possibly indicating this cluster to be an area of active tumour infiltration.

We investigated the expression of other immune cell-marker proteins because of the above differences in immune processes within clusters 6 and 13, and highly localised CD45 abundance and staining (Supplementary Fig. 5f). Two neutrophil markers, neutrophil cytosolic factor 2 & 4 (NCF2 & NCF4), show colocalization with CD45 in the upper left of the sampled region along with two marker proteins for pro-tumour M2 macrophages, CD163 & mannose receptor C-type 1

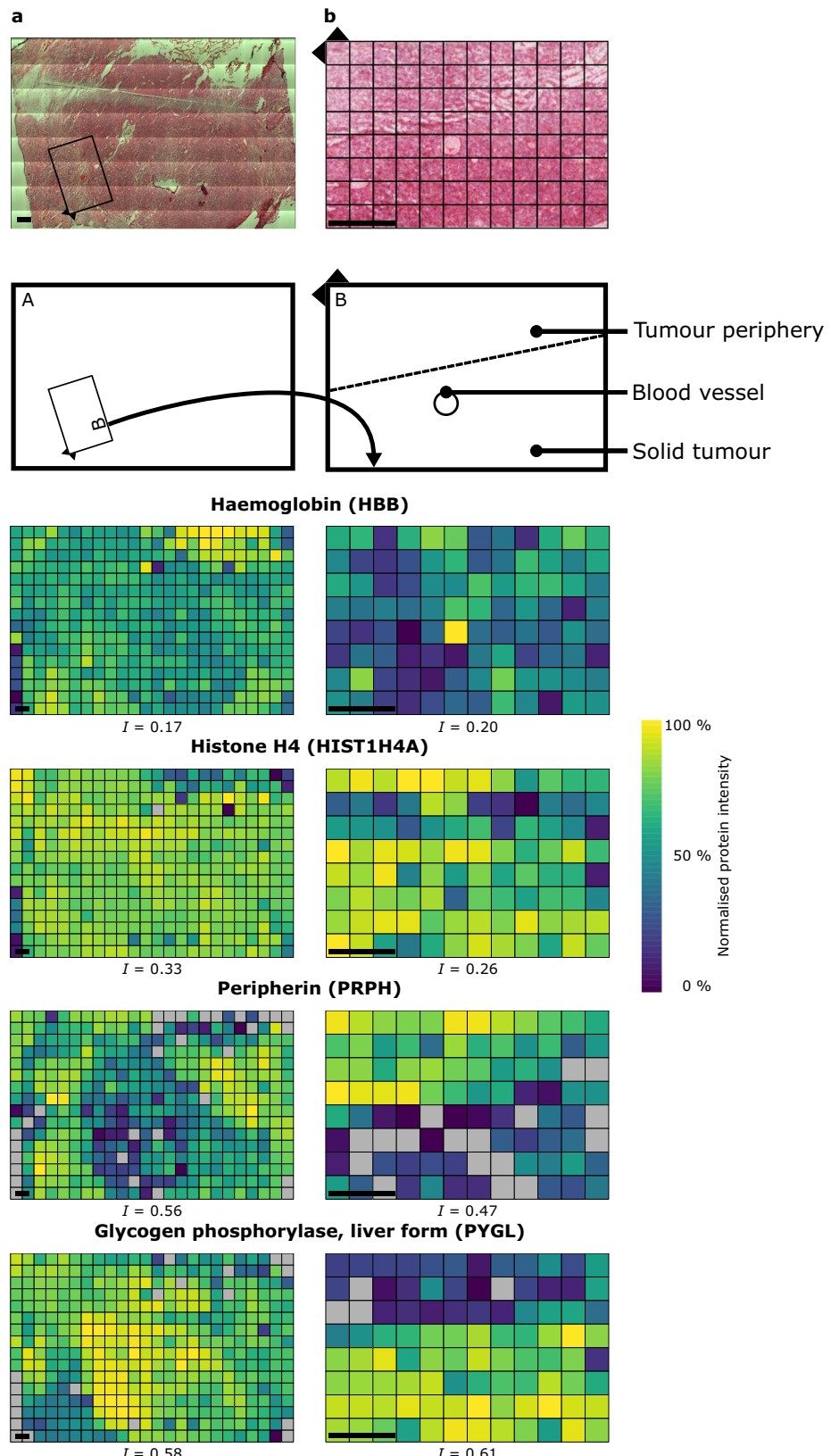

**Fig. 2 | Spatial proteomic maps of AT/RT tumour tissue reveal regional boundaries. a** Normalised protein intensity of four example proteins mapped back to the original spatial positions within the atypical teratoid-rhabdoid tumour (AT/RT) tumour tissue at a resolution of 833 μm ($n = 1$) (**a**) and 350 μm (representative from three independent experiments) (**b**) with their corresponding Moran's Index of spatial autocorrelation (I). Box in **a** represents the area analysed in an adjacent tissue section (**b**). Scale bar = 1 mm. Normalised protein intensities are scaled separately for each protein. Grey = not detected.

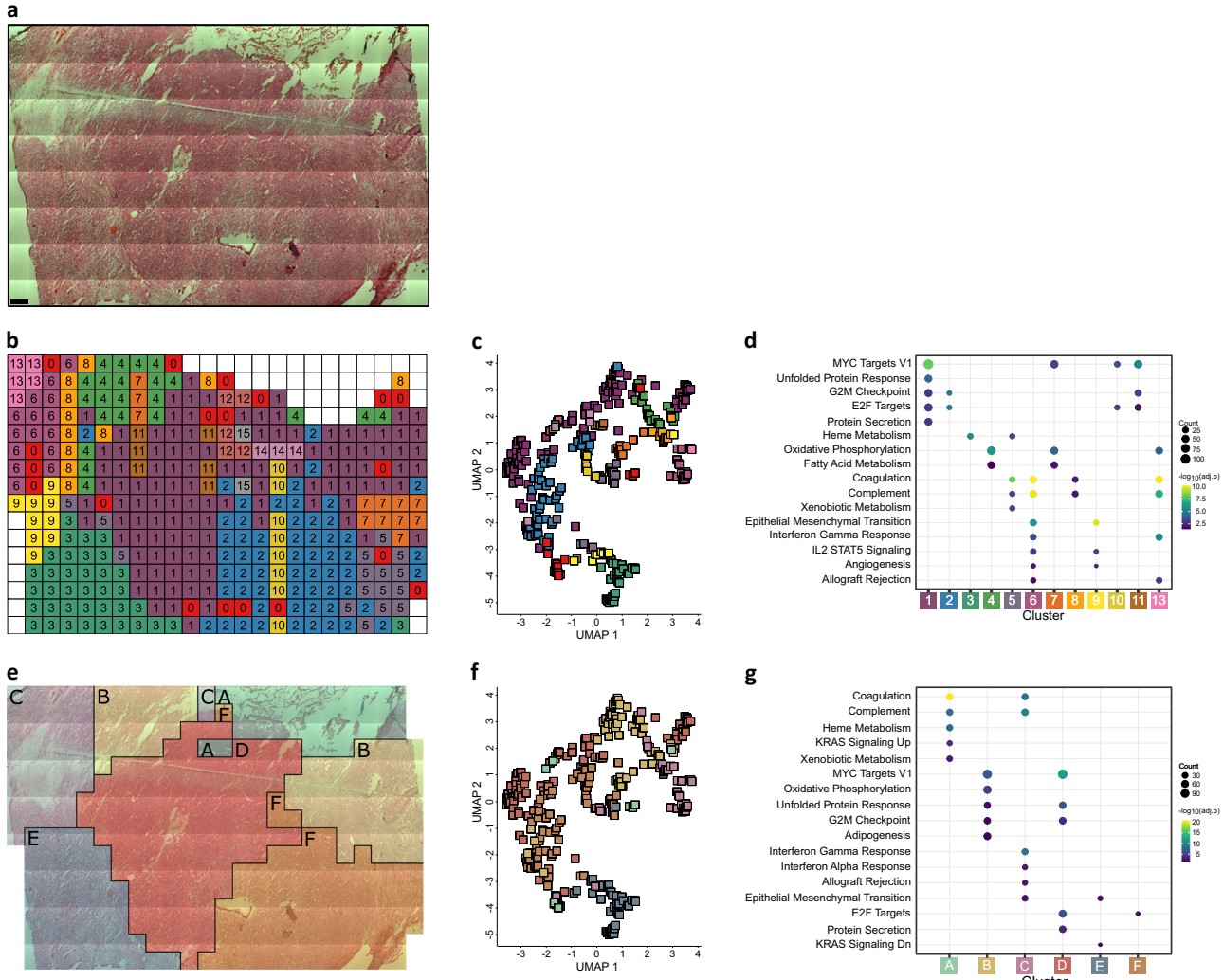

**Fig. 3 | Clustering and functional analysis displays regional AT/RT proteomic maps in the tumour microenvironment at 833 μm spatial resolution. a** H&E Image of AT/RT tissue (*n* = 1). **b** Map of cluster assignment based on hierarchical clustering and the dynamic tree cut algorithm (spatially-unaware). **c** UMAP embedding of data coloured by cluster assignment in **b**. **d** Enriched MSigDB Hallmark gene sets within marker proteins of clusters shown in the cluster map. Enriched MSigDB Hallmark gene sets within marker proteins (two-sided Wilcox test, Benjamini-Hochberg multiple testing correction threshold of 1%) of clusters shown in the cluster map. Significantly enriched hallmarks (one-sided hypergeometric test, Benjamini-Hochberg multiple testing correction threshold of 5%) for each cluster are indicated by the presence of circles. The size and colour of the circles represent the number of proteins contributing to that term and the adjusted *p* value of the enrichment, respectively. **e** Cluster-map of AT/RT tissue generated by Affinity Network Fusion. This clustering is spatially-aware, meaning the relative spatial locations of samples and protein abundance values are used to generate the clusters. **f** UMAP embedding of data coloured by ANF cluster assignment in **e**. **g** Enriched MSigDB Hallmark gene sets within marker proteins of clusters shown in the cluster map generated by ANF, as in **d**. Source data are provided as a Source Data file.

(CD163 & MRC1)[42,43]. These proteins' peak expression locations correspond with clusters 13 and 6 for the neutrophil and macrophage markers, respectively (Supplementary Fig. 6) and shows increased abundance for many proteins involved in neutrophil function and other immune-related processes such as B-cell differentiation and the JNK cascade within cluster 13 (Supplementary Fig. 7). Within cluster 6, proteins involved in cell death, the cell cycle, morphogenesis, hedgehog signalling, collagen, and cytoskeletal organisation show increased abundance (Supplementary Fig. 7).

To discover spatial relationships between phenotypically similar and distinct areas, we performed unbiased spatial clustering by integrating a proteomic similarity matrix (rank correlation of top 25% most variable proteins) with a complementary spatial similarity network (based on Euclidean distances) using Affinity Network Fusion (ANF)[44]. This spatially-aware clustering method resulted in six clusters (Cluster A–F) covering the regions of solid tumour, brain/tumour interface,

immune infiltration, and haemorrhage (Fig. 3e). Overlaying this clustering result onto the UMAP dimensionality reduction again shows that these clusters are generally found near each other in the UMAP plot (Fig. 3f).

As above, the clusters were tested for significant enrichment of MSigDB hallmark gene sets (Fig. 3g). Cluster A shows enrichment for blood-related hallmarks, consistent with this cluster comprising predominantly haemorrhage. Clusters B, D and F show enrichment for cell cycle and growth-related hallmarks, consistent with the 'solid tumour' histology evaluation. Cluster C shows enrichment of immune-related hallmarks and the Epithelial-Mesenchymal Transition hallmark, consistent with the presence of immune cell-marker proteins above. Cluster E shows enrichment for the "KRAS Signalling Dn" and "Epithelial-Mesenchymal Transition" hallmarks.

Clustering in high- and low-resolution maps was broadly consistent (Fig. 4a, b) with generally contiguous clusters that represent the

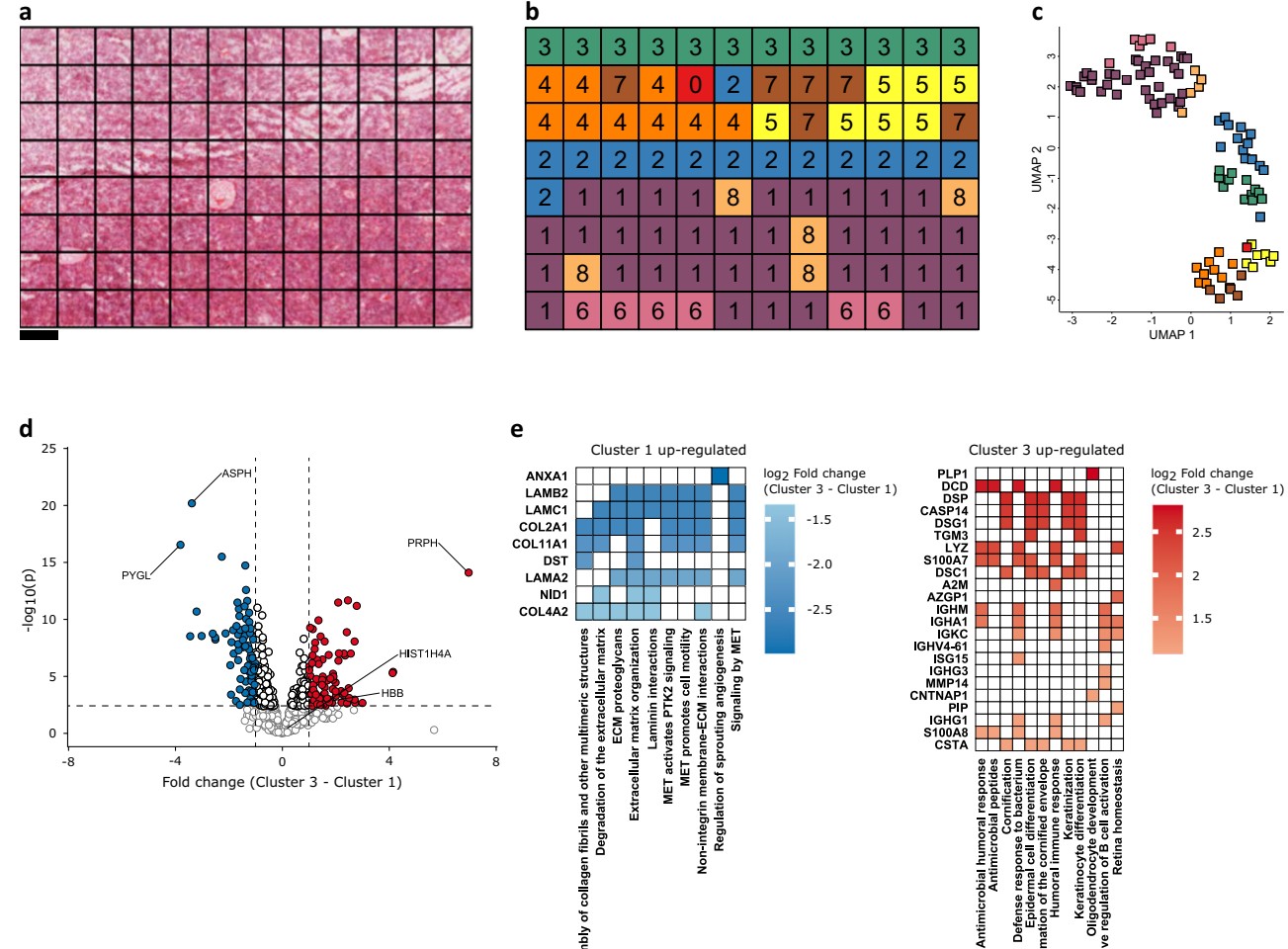

**Fig. 4 | Proteomic map clustering defines functional layers at the tumour periphery at 350 μm spatial resolution. a** H&E image of AT/RT tissue (representative from three independent experiments). **b** Map of cluster assignment based on hierarchical clustering and the dynamic tree cut algorithm (spatially-unaware). Cluster 3 corresponds to brain/tumour interface. Cluster 1 corresponds to solid tumour. Clusters 4,7 & 5 correspond to the transition between solid tumour and brain/tumour interface. **c** UMAP embedding of data coloured by cluster assignment in **b**. **d** Volcano plot of clusters 3 against cluster 1 from **b**. Significance was calculated using a two-sided *t*-test, *p*-values were corrected for multiple testing using Benjamini-Hochberg 5% FDR threshold. The horizontal dashed line represents an FDR of 5%. The vertical dashed line represents −/+ 2-fold-change. **e** GSEA of clusters 1 and 3 from **b**. Gene set membership is indicated by colouring the cell with that protein's log₂ fold-change. Source data are provided as a Source Data file.

solid tumour (cluster 1), the brain/tumour interface (cluster 3), the margin between (clusters 2, 4, 5, and 7), and blood vessels (cluster 8). After UMAP dimensionality reduction, these clusters are again found near each other within the UMAP visualisation (Fig. 4c). A volcano plot between cluster 3 and cluster 1 reflects the previously observed differential abundance of PYGL, ASPH and PRPH and other marker candidates (Fig. 4d). Functional analysis of clusters 1 & 3 indicates that proteins involved in extracellular matrix, cell adhesion & motility, angiogenesis, immune processes, epidermis function, and neuronal development are differentially abundant between solid tumour and brain/tumour interface (Fig. 4e).

### High resolution proteomic maps visualise proteomic patterns around blood vessels

We further focussed on an area containing four blood vessels, represented in a single voxel in the previous analysis (350 μm spatial resolution), allowing us to map protein abundance patterns to potential nutrient/oxygen gradients within the tumour tissue by proxy of distance of individual cells to blood vessels.

This region was subdivided into a 9-by-9 grid, resulting in 40 μm spatial resolution (Fig. 5a). Each 40 μm × 40 μm × 10 μm voxel contained between 5 and 10 visible nuclei. In total, 1550 proteins were quantified using data-independent analysis (DIA-PASEF[45]) on a Bruker timsTOF SCP at a throughput of 40 samples per day (Supplementary Data 3).

We then measured the distance of each cell to the nearest blood vessel (Fig. 5b) to use this distance as a proxy for nutrient and oxygen availability. As expected, we observed positive correlation of blood proteins with closeness to blood vessels (haemoglobin, Fig. 5c)). After spatial clustering as above, the tissue is segmented into four main clusters (Fig. 5d), representing voxels further from blood vessels (cluster 1), and voxels immediately next to or containing blood vessels (clusters 2, 3 & 4). UMAP dimensionality reduction shows general separation between voxels within cluster 1 and other voxels (Supplementary Fig. 8a–c). Testing for enrichment of MSigDB hallmark gene sets within the clusters shows an enrichment of the "Oxidative Phosphorylation" term within cluster 1, and terms related to blood within clusters 2 & 4 (Supplementary Fig. 8d).

GAPDH shows consistent intensity across the tissue (Fig. 5e). Two main patterns are visible in the generated proteomic maps in Fig. 5e: proteins correlated to voxel:blood vessel distance and proteins anticorrelated to voxel:blood vessel distance. Alpha-2-Macroglobulin, a

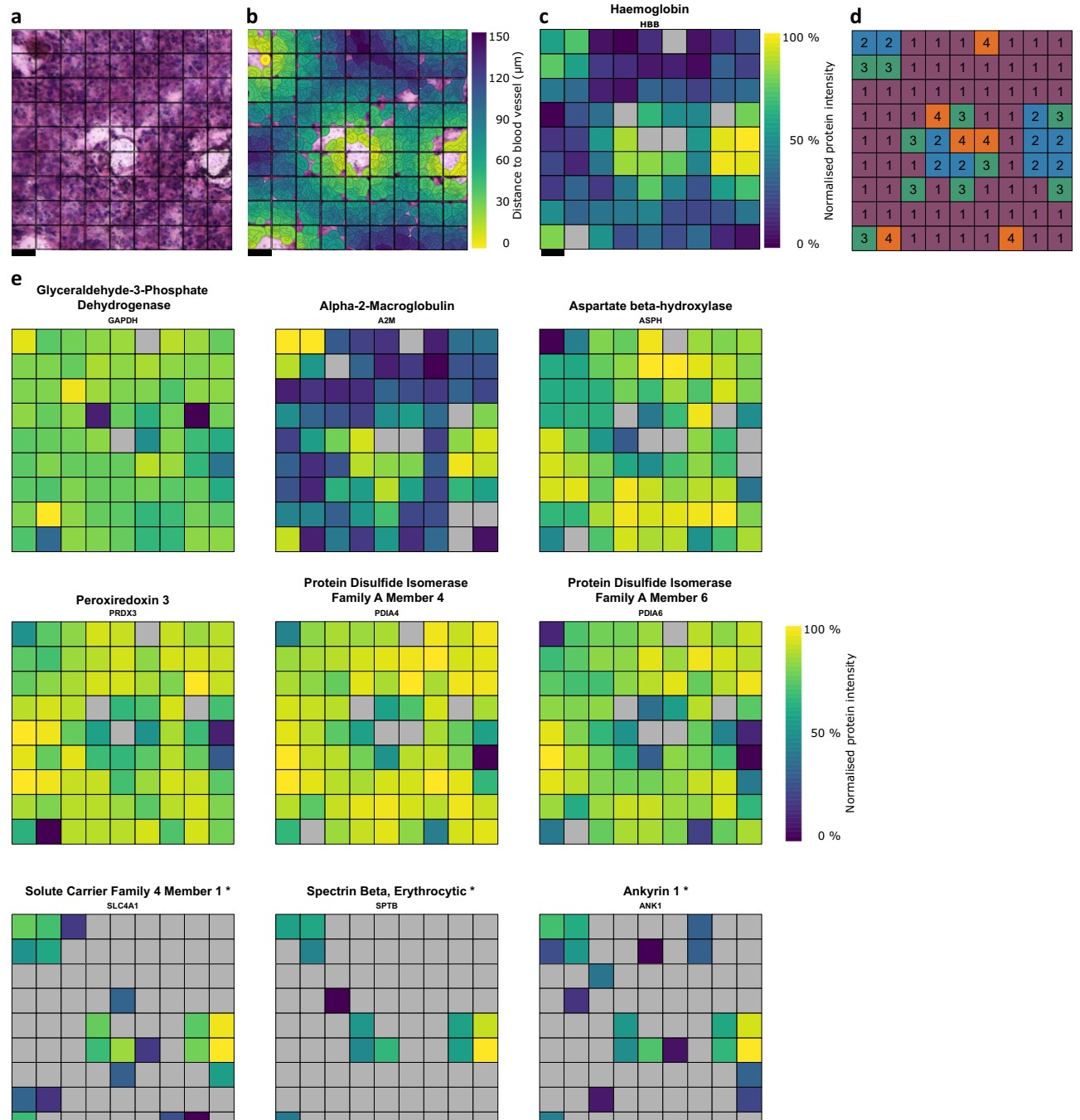

**Fig. 5 | High resolution proteomic maps visualise proteomic patterns around blood vessels at 40 μm spatial resolution. a** H&E Image of AT/RT tissue. Scale bar represents 40 μm. *n* = 1. **b** Measurement map of distance from each cell to the nearest blood vessel from the image in **a**. **c** Proteomic map of haemoglobin distribution within the tissue. **d** Map of cluster assignment based on hierarchical clustering and the dynamic tree cut algorithm (spatially-unaware). **e** Proteomic maps of selected proteins either correlating or anticorrelating with blood vessel location at 40 μm resolution. Erythrocyte markers are denoted with '*'. Normalised protein intensities are scaled separately for each protein. Grey = not detected. Scale bar = 40 μm.

highly abundant blood protein forms an intensity gradient around blood vessels, correlating with haemoglobin. Aspartate beta-hydroxylase (ASPH) shows highest intensity in those voxels furthest away from blood vessels, consistent with observations that ASPH can be regulated by hypoxia[46,47]. Peroxiredoxin 3, suggested to protect cells against apoptosis caused by oxidative stress in hypoxia[48], Protein disulfide isomerase family A members 4 and 6, suggested to contribute to avoidance of cell death pathways in some tumours and activation of Wnt signalling[49,50], also show increased intensity in voxels further away from vessels, indicating the presence of a cancer proteo-phenotype

within the tumour tissue and in dependency of oxygen/nutrient availability in otherwise homogeneous tissue. In addition, three erythrocyte markers show specific, high intensity within voxels containing, or that are very close to blood vessels: Solute carrier family 4 member 1, Spectrin Beta, and Ankyrin 1[51–53].

## Spatial network clustering reveals extracellular matrix and integrin receptor heterogeneity

Throughout the spatial analysis and manual inspection of the data we noticed the common presence of many extracellular matrix-related

proteins in the resulting enriched pathways and highly spatially-correlated proteins. Because of the relevance of the ECM for tumour development and infiltration into healthy brain, we performed ANF clustering based on the proteins annotated as 'Core Matrisome' proteins within MatrisomeDB[54]. This resulted in nine clusters (A-I) which are generally spatially contiguous across the tissue (Fig. 6a). ECM-defined clusters were used for functional enrichment of proteins within each cluster, (Fig. 6b) and are generally consistent with the functional enrichments in Fig. 4e without ECM focus, linking overall spatial proteome distribution with ECM architecture in the tumour structure. To determine whether differential ECM abundance or composition was driving this clustering, we plotted the summed (Supplementary Fig. 9a) and mean (Supplementary Fig. 9b) abundance of the core matrisome proteins. These aggregate ECM spatial abundances show that both total ECM abundance (in the haemorrhage and immune infiltration area; clusters A and C in Fig. 6a) and ECM composition (in the remaining clusters) are contributing to these cluster definitions.

Plotting the spatial distributions of detected collagens reveals broad heterogeneity in abundance of collagen proteins across the different collagen subfamilies (Fig. 6c). The fibrillar collagens COL2A1, COL11A1, and COL11A2 show higher expression in the solid tumour region and other fibrillar collagens, COL1A2 and COL3A1, show higher abundance in the brain/tumour interface region. Other collagen subfamily members also show differential spatial abundance: COL6A1, COL6A2, COL26A1 (filament forming collagens); COL12A1, COL14A1 (fibril associated collagens). Collagens 4A1, 4A2 (network forming collagens), 15A1 and 18A1 (multiplexins) show spatially homogeneous expression. These spatial distributions broadly map to the ANF clusters. In addition, several integrin proteins also show spatial heterogeneity within the tissue (Fig. 6c). Integrin α5, α7, α10, αV, αM and β2 show spatial heterogeneity whereas integrin β1 shows relatively homogeneous expression. Other proteins associated with the ECM also show spatial heterogeneity such as CD44, Lysyl oxidase (LOX), Lumican (LUM) (lower in solid tumour region), Versican (VCAN), and Cathepsin D & G (higher in immune cell-enriched region). In general ECM and ECM-associated proteins as defined in MatrisomeDB show a higher mean Moran's I value than non-ECM proteins (ECM: I = 0.334, ECM-associated: I = 0.320, non-ECM: I = 0.219) (Supplementary Fig. 10)[47].

## Discussion

Spatial proteomics is a rapidly developing research area that aims to provide insights into health and disease within the spatial context of biological macrostructures such as organs, tissues, or tumours. The use of spatial technologies in disease research can help to understand drug action and delivery, the activities of the immune system, all while preserving critical location information[55]. Spatial transcriptomics studies such as Zheng et al. have started using machine learning to derive diagnostic and prognostic value in several oncology contexts, indicating that spatial 'omics data can aid pathological assessment of tumour tissue at the point of care[56–60]. Integration of this approach with other 'omics modalities such as proteomics, lipidomics and metabolomics has the potential to further refine the acquisition of mechanistic information about an individual's pathology. To date, spatial proteomics has largely been used to visualise and overlay proteome information on existing, discernible features after staining or targeted detection of markers[22–26], or to compare features through their proteome such as tumour and peripheral tissue[21]. However, such approaches can introduce confirmation bias, preventing the detection of novel spatial features that may not be readily apparent by a feature-driven detection. The spatial proteomics workflow presented here addresses the need for highly multiplexed, quantitative, spatially-resolved, systematic measurements of proteins within tissue to understand the spatial organisation of molecular pathways in health and disease.

Our methods involve the systematic sampling of tissue subsections using laser capture microdissection (LCM), sample preparation, liquid chromatography-mass spectrometry and advanced statistical analysis of topographic data, using widely available equipment. We have demonstrated that these optimised methods can detect proteins and pathways with spatially variable abundance within both tumour and tumour-infiltrated normal tissue without introduction of sampling bias. Critically, our spatially-aware data analysis enables the identification of processes with deep molecular resolution without prior knowledge of the tissue composition, thus creating an objective, unbiased approach to deep phenotyping pathological tissue in its biological context and discovering features, such as areas of heightened immune response.

Currently, an increasing number of studies are focused on feature-driven LCM-coupled proteomics. In contrast, we propose a systematic approach that allows for the unbiased generation of comprehensive proteomic maps at the individual protein and pathway level, driven exclusively by the acquired proteome data. These maps can fulfil the requirements for feature-driven analysis by reconstituting features from systematic sampling and allow the discovery of new proteo-phenotypes without the need for visually identifiable parameters.

Through our generation of over 5000 proteomic maps, we have demonstrated the presence of molecular heterogeneity at multiple scales within AT/RT tumour tissue sections, revealing proteomic differences between areas of tissue that appear visually homogeneous with good agreement with immunohistochemical orthogonal confirmation of protein expression. Our approach provides information on immune cell infiltration and state within the tissue by detecting neutrophil and pro-tumour M2 macrophage markers at different distances from the solid tumour; it will be interesting to correlate these proteome-derived features with those of spatial or single-cell transcriptomics. AT/RTs have a low mutational burden and show a range of programmed death-ligand 1 (PD-L1) expression, mainly low-medium expression, and the SHH subtype generally shows a low level of immune infiltration[61–63]. Our data is largely consistent with this. However, we observed a focussed immune response in a small spatially defined area in the tumour periphery, giving insights into the pathology of AT/RT. The identification of this area was likely possible by our selection of a large tumour block (most AT/RTs of the brain are sampled only as small needle biopsies). An ongoing immune response at the edge of the neoplasm even in later stages of the disease support the idea that future immunotherapy may be a valuable adjunct to current, largely untargeted therapies[61,64,65].

Interestingly, a recent publication by Paassen et al. has demonstrated that a subset of the SHH AT/RT-subtype are sensitive to NOTCH inhibition when cultured as tumouroids[66]. We show that the region of solid tumour has a high intensity of aspartate beta-hydroxylase (ASPH), which is an activator of the NOTCH signalling pathway[67].

We also have demonstrated that our unbiased spatially-resolved proteomics approach provides data on the tumour microenvironment. The increased level of fibrillar collagens within the solid tumour is consistent with observations of increased fibrillar collagen deposition in glioblastomas and the measurement of increased tissue stiffness of tumours, including brain tumours[68,69]. Additionally, the spatial abundance of integrin subunits αM and β2 correlates with neutrophil marker abundance, strongly suggesting localised neutrophil accumulation[70]. Furthermore, integrin α10 abundance correlates well with the abundance of COL2A1 and COL11A1, consistent with the observation that the α10β1 integrin receptor binds to type 2 and type 11 collagens[71]. Although functional interaction between these proteins cannot be determined from the methods used here, further investigation into integrin receptor signalling in AT/RT could be a worthwhile

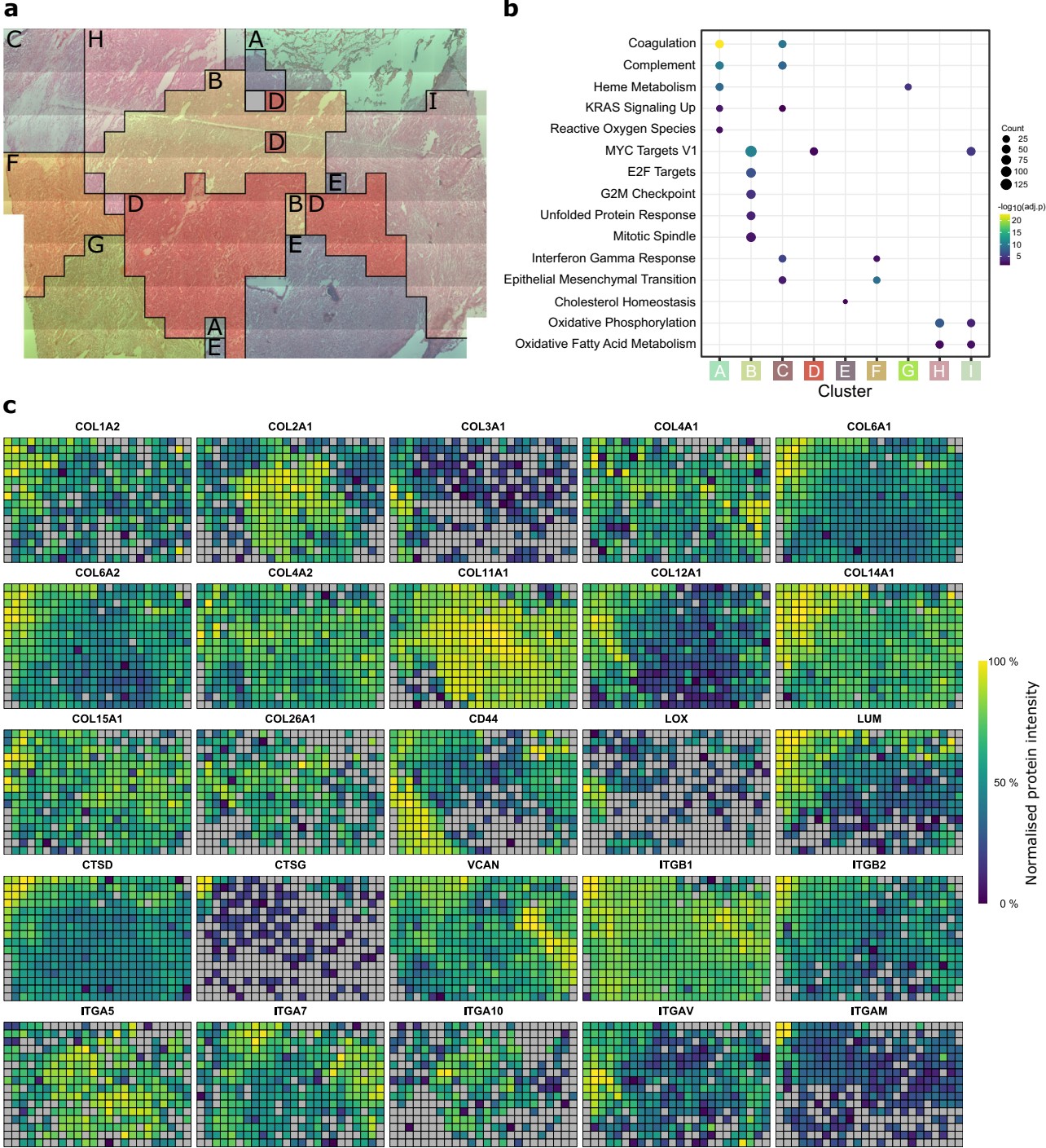

**Fig. 6 | Spatial heterogeneity of extracellular matrix proteins across the AT/RT tumour at 833 μm spatial resolution. a** Cluster-map of AT/RT tissue generated by affinity network fusion (ANF) of core matrisome proteins as defined in MatrisomeDB. This clustering is spatially-aware, meaning the relative spatial locations of samples and protein abundance values are used to generate these clusters. The grey voxel represents one sample where no core matrisome proteins were detected. **b** Enriched MSigDB Hallmark gene sets within marker proteins (two-sided Wilcox test, Benjamini-Hochberg multiple testing correction threshold of 1%) of clusters shown in the cluster map. Significantly enriched hallmarks (one-sided hypergeometric test, Benjamini-Hochberg multiple testing correction threshold of 5%) for each cluster are indicated by the presence of circles. The size and colour of the circles represent the number of proteins contributing to that term and the adjusted *p* value of the enrichment, respectively. **c** Proteomic maps of selected extracellular matrix proteins and integrin receptors at 833 μm resolution. Normalised protein intensities are scaled separately for each protein. Grey = not detected.

avenue of investigation as integrin α10/β1 has shown potential as a therapeutic target in other malignant primary brain tumours[72].

Further, we demonstrate that our unbiased tumour proteome can identify molecular gradients associated with nutrient availability relative to the distance of tumour cells from a (medium-sized) blood vessel. This can be detected in otherwise morphologically homogeneous tissue and, as far as diffusible or extracellularly deposited molecules are concerned, would be difficult to infer from transcriptomic studies, as transcription site and eventual localisation of a protein product are usually not congruent. This sets the scene for

future integrated spatially-resolved transcriptomic and proteomic studies, which will provide a more complete understanding of disease processes by using advanced bioinformatics tools[73,74]. Interpretation of such data will benefit from further development of analytical and statistical methods that are aware of spatial relationships between samples. This could include using missing value imputation and machine learning approaches[75,76].

A limitation of our study is that there is the requirement to strike a balance between the need for spatial resolution and the need for meaningful depth to cover pathways of interest. These limiting factors are driven by technical considerations such as sensitivity and throughput of the LC-MS systems used. Increasing the spatial resolution demonstrated here, towards the single-cell level and covering comparable areas, poses a formidable analytical challenge, which further escalates when analysing tissue in three dimensions. Several recent studies have developed methods towards increased sensitivity and throughput of low-abundance samples. Kreimer et al. demonstrate an efficient dual trap-column liquid chromatography configuration for the analysis of single cells, and Derks et al. and Thielert et al. have both demonstrated low multiplexed methods of data independent analysis, with both frameworks offering the potential for future increases in multiplexing[77–79]. The similar analytical demands of systematic spatial proteomics analysis and single cell proteomics will allow both fields to develop in lock-step. Furthermore, the current trend in commercial mass spectrometry for proteomics towards faster, more sensitive instruments will also likely provide benefits[80,81]. Despite these developments and those of novel high-throughput LC-MS platforms, that can now robustly analyse 1000 s of samples relatively quickly[82–84], future approaches are likely to use systematic spatial proteomic analysis, possibly compromising spatial resolution but incorporating elements of machine learning, as can be done on spatial transcriptomics data[56,58,85–87]. We believe that the high depth of our detected proteomes in spatially sampled LCM-derived tissue will be highly complementary to current tissue imaging technologies such as MALDI-MS where spatial resolution may be higher but proteome depth much shallower. A combination of these approaches has great potential to increase our understanding of spatially-resolved biological and pathological processes in human tissue at the molecular level[24,33,88].

## Methods

### Tissue retrieval and processing
Post-mortem brain tissue was retrieved by the Oxford Brain Bank; a research ethics committee (REC) approved and Human Tissue Authority (HTA) -regulated research tissue bank (REC reference 15/SC/0639, issued by the NHS Health Research Authority 'South-Central – Oxford C'). Tissue was donated and analysed after full written consent was obtained from the next of kin of the tissue donor. The Oxford Brain Bank is hosted by the Nuffield Department of Clinical Neurosciences, University of Oxford. Sex and gender were not considered in the study design due to the case study nature. The retrieved brain was sectioned into 1 cm thick coronal sections starting at the level of the mammillary bodies. Whole hemisphere slabs were snap-frozen in liquid nitrogen vapour (to minimise freezing artefacts) and stored at −80 °C until further dissection on dry ice. Due to its large size, tumour tissue was present within multiple of these coronal slices. The tumour tissue was dissected from the first coronal and second posterior coronal slices (P1 and P2). The tumour from the P2 slice was split into quadrants. Cryosections were taken from all pieces to determine the tissue block with the best morphological and cellular preservation. Cryosections were stained with H&E (see below) and examined by a Neuropathologist (OA). The upper-right quadrant from the P2 coronal slice was selected for use in further experiments. Details of the tumour phenotype and genotype are provided in Supplementary Fig. 1.

Relevant tissue blocks of the AT/RT tumour were acclimatised to −20 °C and mounted onto a cryostat block using OCT Compound (Cell

Path, ARG1180). Careful consideration was taken to ensure cut sections were not contaminated with OCT. Sections were cut at 10 μm and mounted onto UV irradiated (254 nm, 30 minutes) 1.0 PEN membrane slides (Zeiss) at −18 °C for LCM or Superfrost glass slides for histology. Sections were then air-dried for several minutes and placed onto a Shandon Linistain for automated H&E staining. Sections were fixed in 70% denatured alcohol, hydrated, stained with Harris' Haematoxylin, incubated in 0.4% acid alcohol, placed in Scot's tap water, and stained with Eosin containing 0.25% acetic acid with regular washing steps in between. Stained sections were then dehydrated in increasing concentrations of denatured alcohol and air-dried without coverslips and stored at −80 °C until processing by laser-capture microdissection.

### Laser capture microdissection
Areas of tissue analysed were annotated and isolated from the prepared slides using a laser-capture microscope equipped with laser pressure catapulting (PALM Microbeam, Zeiss). Cutting and capturing the annotated tissue areas were performed automatically and used the 10x objective lens. The settings in the control software for cutting were Energy: 43, Focus: 55; and for capturing were Energy 20, Focus −15. Samples were collected into 20 μL RIPA buffer (Pierce #89900) in the cap of 200 μL PCR tubes or PCR-cap strips of 8. Collected samples were immediately placed in dry ice. Samples were stored at −80 °C until further use.

For the 40 μm resolution data, tissue areas were microdissected using a Leica LMD7 laser capture microscope. The region of interest was imaged using a 20x Objective lens with a 10% tile overlap using the LASX software (Leica). Image tiles were stitched, then imported into QuPath version 0.4.3[89]. Voxel annotations were created manually in a 9-by-9 grid with each voxel having a side-length of 40 μm, and three reference points for image to stage location registration were selected. Microdissection coordinates were then exported from QuPath via a custom plug-in and imported into the LMD7 software[90]. The following settings were used: Power 50, aperture 2, speed 10, middle pulse count 1, head current 90%, pulse frequency 500. Tissue areas were cut into wells of a 96-well PCR plate. To ensure the collected tissue was located at the bottom of the wells, 100 μL of acetonitrile was added to each well, the plate was then sealed, centrifuged at 1000 x $g$ for 5 minutes, then dried in a vacuum centrifuge at 40 °C until dryness. The plate was then stored at −20 °C until further processing.

### Proteomic sample processing
Samples were thawed, incubated at room temperature for 30 minutes and briefly centrifuged. Caps were rinsed with 20 μL of RIPA buffer (#89900, Pierce) containing 25 units of Benzonase (E1014, Merck) to collect any remaining tissue and briefly centrifuged, followed by incubation at room temperature for 30 minutes to degrade DNA and RNA. Proteins were reduced by adding DTT to 5 mM and incubated at room temperature for 30 minutes, followed by the addition of iodoacetamide to 20 mM and incubation at room temperature for 30 minutes.

Paramagnetic SP3 beads (GE45152105050250 & GE65152105050250, Cytiva) were prepared as described by Hughes et al. and processed by a modified SP3 protocol[24,91,92]. Three μL of SP3 beads were mixed with the samples, and acetonitrile added to a final concentration of 70% (v/v). Samples were mixed with 1000 rpm orbital shaking for 18 minutes, followed by bead immobilisation on a magnet for 2 minutes. The supernatant was discarded, and beads were washed twice with 70% (v/v) ethanol in water and once with 100% acetonitrile without removal from the magnet. Beads were resuspended in 50 mM ammonium bicarbonate containing 25 ng of Trypsin (V5111, Promega) and digested overnight at 37 °C. After digestion, the beads were resuspended by bath sonication. Acetonitrile was added to the samples to 95% (v/v) and shaken at 1000 rpm for 18 minutes. Beads were immobilised on a magnet for 2 minutes, and the supernatant

discarded. Beads were resuspended in 2% acetonitrile and immobilised on a magnet for 5 minutes. Peptides were transferred to glass LC-MS vials or 96-well PCR plates containing formic acid in water, resulting in a final formic acid concentration of 0.1%.

For the 40 μm resolution data, a single step digestion was performed. To the dried samples, 4 μL of digestion buffer was added, the plate sealed and incubated at 50 °C for 90 minutes in a thermocycler[93]. The digestion buffer contained: 0.2% n-dodecyl-β-D-maltoside (DDM), 1 ng/μL Trypsin/LysC Mix (Promega), 100 mM triethylammonium bicarbonate. After incubation, the plate was cooled to 20 °C then removed from the thermocycler. The resulting peptides were loaded onto Evotip Pure C18 tips (EvoSep) following the manufacturer's protocol for analysis by LC-MS/MS.

## LC-MS/MS

Peptides from 833 μm resolution samples were analysed by LC-MS/MS using a Dionex Ultimate 3000 (Thermo Scientific) coupled to a timsTOF Pro (Bruker) using a 75 μm x 150 mm C18 column with 1.6 μm particles (IonOpticks) at a flow rate of 400 nL/min. A 17-minute linear gradient from 2% buffer B to 30% buffer B (A: 0.1% formic acid in water. B: 0.1% formic acid in acetonitrile) was used[94]. The TimsTOF Pro was operated in PASEF mode. The TIMS accumulation and ramp times were set to 100 ms, and mass spectra were recorded from 100–1700 m/z, with a 0.85–1.30 Vs/cm$^2$ ion mobility range. Precursors were selected for fragmentation from an area of the full TIMS-MS scan that excludes most ions with a charge state of 1 +. Those selected precursors were isolated with an ion mobility dependent collision energy, increasing linearly from 27–45 eV over the ion mobility range. Three PASEF MS/MS scans were collected per full TIMS-MS scan, giving a duty cycle of 0.53 s. Ions were included in the PASEF MS/MS scan if they met the target intensity threshold of 2000 and were sampled multiple times until a summed target intensity of 10000 was reached. A dynamic exclusion window of 0.015 m/z by 0.015 Vs/cm$^2$ was used, and sampled ions were excluded from reanalysis for 24 seconds.

Peptides from 350 μm resolution samples were analysed by nano-UPLC-MS/MS using a Dionex Ultimate 3000 coupled to an Orbitrap Fusion Lumos (Thermo Scientific) using a 75 μm x 500 mm C18 EASY-Spray Columns with 2 μm particles (Thermo Scientific) at a flow rate of 250 nL/min. A 60-minute linear gradient from 2% buffer B to 35% buffer B (A: 5% DMSO, 0.1% formic acid in water. B: 5% DMSO, 0.1% formic acid in acetonitrile). MS1 scans were acquired in the Orbitrap between 400 and 1500 m/z with a resolution of 120,000 and an AGC target of $4 \times 10^5$. Precursor ions between charge state 2+ and 7+ and above the intensity threshold of $5 \times 10^3$ were selected for HCD fragmentation at a normalised collision energy of 28%, an AGC target of $4 \times 10^3$, a maximum injection time of 80 ms and a dynamic exclusion window of 30 s. MS/MS spectra were acquired in the ion trap using the rapid scan mode.

Peptides from 40 μm resolution samples were analysed using an Evosep One LC system (EvoSep) coupled to a timsTOF SCP mass spectrometer (Bruker) using the Whisper 40 samples per day method and a 75 μm x 150 mm C18 column with 1.7 μm particles and an integrated Captive Spray Emitter (IonOpticks). Buffer A was 0.1% formic acid in water, Buffer B was 0.1% formic acid in acetonitrile. Data was collected using diaPASEF[45] with 1 MS frame and 9 diaPASEF frames per cycle with an accumulation and ramp time of 100 ms, for a total cycle time of 1.07 seconds. The diaPASEF frames were separated into 3 ion mobility windows, in total covering the 400 – 1000 m/z mass range with 25 m/z-wide windows between an ion mobility range of 0.64–1.4 Vs/cm$^2$. The collision energy was ramped linearly over the ion mobility range, with 20 eV applied at 0.6 Vs/cm$^2$ to 59 eV at 1.6 Vs/cm$^2$.

## Proteomic data analysis

For the 833 μm and 350 μm resolution data, raw data files were searched against the UniProtKB human database (Retrieved 17/01/2017, 92527 sequences) using MaxQuant version 1.6.14.0, allowing for tryptic specificity with up to 2 missed cleavages. Cysteine carbamidomethylation was set as a fixed modification. Methionine oxidation and protein N-terminal acetylation were set as variable modifications and the "match between runs (MBR)" option was used (MBR was not used for tissue titration data). All other settings were left as default. Label-free quantification was performed using the MaxLFQ algorithm within MaxQuant[95,96]. Protein and peptide false discovery rate (FDR) levels were set to 1%.

For the 40 μm resolution data, raw files were analysed in DIA-NN[97] version 1.8.1 using an *in-silico* spectral library generated by DIA-NN with default settings (1 missed cleavage, N-terminal methionine excision was allowed) using a Uniprot human FASTA file containing 20383 reviewed sequences. MS1 and MS2 accuracies were set to 15 ppm, all other settings were left as default.

The MaxQuant and DIA-NN output files containing the protein-level information are included as supplementary data files 1, 2, and 3 for the 833, 350 and 40 μm resolution data, respectively.

## Spatial data analysis

The mapping of the mass spectrometry raw files to their relative spatial locations is shown in Supplementary Data 4. The spatial analysis uses functions within the spdep and raster R packages[98–100]. MaxQuant's protein level output files ('proteingroups.txt') were filtered to remove reverse hits, 'Only identified by site' hits and potential contaminants. The 'LFQ intensity' columns were log$_2$ transformed and then normalised by median subtraction. Protein groups that did not meet a cut-off of having at least 9 voxels with normalised LFQ values are not taken forward for further analysis. The following steps occur independently for each protein group. Normalised LFQ intensities were then coerced into a matrix reflecting the rastered pattern of sample acquisition. The quantification matrix was converted into a raster object and then to a polygon object using the raster R package. From this polygon object, a neighbour list was built for each voxel of the raster using the 'Queen's Case' where cells are considered neighbours if they share an edge or a vertex. The neighbour list was then supplemented with a spatial weights matrix using a binary coding scheme where neighbours are given a weighting of '1' and non-neighbours a weighting of '0' in the spatial weights matrix. The raster object and the weighted neighbour list were then used as inputs to a permutation test for the Moran's $I$ statistic, calculated using 999 random spatial permutations of the raster object to calculate pseudo-$p$-values. Moran's $I$ statistics and the associated $p$-values are collected for every protein group. The $p$-values were then corrected for multiple testing using the Benjamini-Hochberg FDR method.

## Immunohistochemistry

Sections were cut as above and were mounted to Superfrost glass slides for IHC and air-dried. Slides were fixed in ice-cold acetone for 10 minutes, washed twice with TBS/T (20 mM Tris, 150 mM NaCl, 0.05% Tween 20) and blocked with 10% goat serum in TBS/T for 60 minutes at room temperature. Primary antibodies were diluted in 5% goat serum in TBS/T and incubated at RT for 60 minutes or 4 °C overnight. Sections were washed three times with TBS/T. Staining visualisation was performed by incubating with a cocktail of anti-mouse and anti-rabbit secondary antibodies conjugated to horse-radish peroxidase (Envision Kit, Agilent) for 60 minutes at room temperature. Sections were then washed with TBS/T three times and incubated with 2% 3,3'-diaminobenzidine for 5 minutes, immersed in water and then counterstained with Harris' Haematoxylin for 1 minute. Primary antibodies used and dilutions: rabbit anti-PYGL, 1:100, 4 °C overnight, HPA000962 (Atlas Antibodies); rabbit anti-ASPH, 1:1000, RT 60 minutes, NBP2-34125 (Novus Biologicals); mouse anti-CD45 (PD7/26 + 2B11), 1:200, RT 60 minutes, ab781 (Abcam).

## Clustering

Empty voxels and voxels covering the large region of haemorrhage were not included. A distance matrix was built containing the Euclidean distance between each voxel's set of protein LFQ values. Hierarchical clustering of the distance matrix was performed in R using the "average" agglomeration method. Dendrograms were cut using the Dynamic Tree Cut method at a height setting of 100[101].

A complete input matrix is required for UMAP visualisation[102], so proteins with fewer than 70% valid values across the experiment were removed. The remaining missing values were imputed in on a per-sample basis by random draws from a normal distribution using a width of 0.3 and a downshift of 1.8. UMAP dimensionality reduction was performed on this imputed data with default settings, and the first two embedding components plotted, and samples coloured according to their cluster assigned by Dynamic Tree Cut at a height of 100.

## Pathway analysis

A two-sample Wilcoxon Rank Sum Test was performed in R for each cluster versus all other clusters to determine marker proteins for each cluster, the p-values were then corrected for multiple testing using the Benjamini-Hochberg FDR method at a 1% threshold. The resulting lists of marker proteins per cluster were used as input to ClusterProfiler's 'compareCluster' function to test for overrepresentation of terms using the MSigDB Hallmark Gene Sets. The entire set of proteins detected in the experiment was used as the background set[103].

## Affinity network fusion

For the network fusion approach, we used a minimally processed expression matrix, log2 transformed and median centred, removing all empty voxels. We selected the top 25% of proteins by variance across all voxels and used this to calculate a proteomic similarity matrix of Spearman's rank correlation coefficients. For the focussed extracellular matrix protein analysis, all detected proteins annotated as "Core Matrisome" in MatrisomeDB were used to calculate the protein similarity matrix. We converted this to a proteomic distance matrix (by taking 1-similarity). Separately, we created a complementary spatial distance matrix representing the Euclidean distance from each voxel location to each other voxel location (where horizontally and vertically adjacent neighbours are distance 1, diagonal neighbours are distance √2 and so on). We converted both matrices into affinity matrices and fused them using Affinity Network Fusion[41]. We then performed spectral clustering on the fused affinity matrix, where the number of clusters (6) was selected using the maximal eigengap heuristic.

## Reporting summary

Further information on research design is available in the Nature Portfolio Reporting Summary linked to this article.

## Data availability

The raw and processed mass spectrometry proteomics data have been deposited to ProteomeXchange Consortium via the PRIDE partner repository. The 833 μm resolution data can be found at PXD039159. The 350 μm resolution data can be found at PXD039398. The 40 μm resolution data can be found at PXD044714. Source data are provided with this paper Source data are provided with this paper.

## Code availability

The R code used for analysis can be downloaded from https://zenodo.org/record/8341909[104]. Code for the affinity network fusion analysis can be accessed at https://github.com/pdcharles/spatial-proteomics-anf.

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

## Acknowledgements

SD acknowledges support from the Nuffield Department of Medicine. SD, PDC, BMK & RF acknowledge support from the Chinese Academy of Medical Sciences Medical Sciences 2018-I2M-2-002. PDC was supported by Pfizer funding awarded to BMK. We acknowledge the Oxford Brain Bank, supported by the Medical Research Council (MRC, MR/L022656/1) and Brains for Dementia Research (BDR) (Alzheimer Society and Alzheimer Research UK). This research project was funded by the NIHR Oxford Biomedical Research Centre (to OA, BRC-1215-20008). The views expressed are those of the authors and not necessarily those of the NHS, the NIHR, or the Department of Health. This work uses data provided by patients and collected by the NHS as part of their care and support and would not have been possible without access to this data. The NIHR recognises and values the role of patient data, securely

accessed, and stored, both in underpinning and leading to improvements in research and care.

## Author contributions

R.F., S.D. and O.A. conceptualized the study. S.D., C.S. and J.O. conducted experiments. R.F., S.D. and P.D.C. analysed data. O.A., R.F. and B.M.K. supervised the experimental work. Funding was acquired by R.F. and O.A. All authors wrote and approved the paper.

## Competing interests

JO is an employee of Bruker Daltonics GmbH & Co. KG. All other authors have no competing interests.
