## [Peer Review File · Nature Communications]

Reviewers' comments:

Reviewer #2 (Remarks to the Author): Expert in AT/RT genomics, clinical research, and tumour microenvironment

Davis et al. report on a workflow for spatially-resolved, quantitative proteomics. As I am not an expert on the technical parameters of spatial proteomics, I comment on the ATRT part. If this manuscript aims to offer new insights on ATRT, I have some questions/comments, which needs to be addressed.

1. The genetic and histologic hallmark of ATRT is the loss of SMARCB1. Please provide a SMARCB1 staining. How does this protein behave at tumor versus non-tumor regions?
2. ATRT are subdivided into three molecular distinct groups (ATR-TYR, ATRT-SHH, ATRT-MYC). Into which subgroup falls this tumor (by methylation arrays).
3. ATRT are known to have some heterogeneity on histopathological level. It would be interesting to see how the heterogeneity in protein expression is related to this histological heterogeneity.
4. The authors report on differences in CD45 protein expression within the sample. Also this result could be easily confirmed by a staining.
5. General concern: the regions which are analyzed with this technic are huge. Multiple cells (hundreds of cells) might be detected within one region. Might the authors comment on what can be the advantage of this platform in comparison to many other platforms on the market, which offer spatial protein analyses with a higher resolution.

Reviewer #3 (Remarks to the Author): Expert in brain cancer genomics, tumour microenvironment, and laser capture microdissection

The present study is original and well-conducted. The authors present a workflow for spatially-resolved, quantitative proteomics of tissue that generates maps of protein abundance across tumor tissue sections and employs spatially-aware algorithms that do not require prior knowledge of the fine tissue structure to detect proteins and pathways with spatial abundance patterns. I am a strong believer in such approaches that tried to diverge from what we are used to seeing and try to identify new information based on the intrinsic molecular characteristics of the tissue. I believe strongly that many of the answers we are seeking are in the tissues section, which represents snapshots of the tumor behavior in vivo. This work represents an attempt to look for these answers utilizing an approach that is innovative in its combination of well know approaches.

The paper needs some modifications. It seems generally that there was a lack of care in presenting the figures that interfered with clarity.

- 1) rephrase sentences 34,35,36,37. It is not clear.
- 2) Figure 1 needs more details to be understood, needs a,b,c letters with explanations in the legend.
- 3) Figure S2 is not explained clearly
- 3) Sentences from 65 to 70: why did you choose these genes PYGL, HBB PRPH, and H4? Put them in the same order as in the figure. Please write the abbreviation of the genes also in the figure
- 4) Figure S3 seems not the right one according to the text.
- 5) Figure 3 needs better referring in the text and authors should point out that the histological images are close-ups of the region of the proteomic map
- 6) Figure 4a needs to be subdivided into sub-panels for better clarity. Authors take things too much for granted and the reader could get lost if the figures are not well explained. Draw some arrows to highlight boundaries
- 7) Figure S4 needs H&E images for better clarity
- 8) The Maldi Imaging section again is not well explained through Figure S6, which is not clear and it seems not to add more information

The approach presented in the paper seems sure laborious, however, if any research group can have access to the technology proposed, it has a strong potential for identifying new leads in cancer worth it to pursue. It seems very demanding to apply this approach to many tissue sections across cohorts of patients-derived tissues. Authors should highlight this limitation and how they would envision its application for identifying robust targets. It is more of a technical paper since it does not leave so far useful scientific information that could be used considering that it is done on one single tissue section.

Reviewer #4 (Remarks to the Author): Expert in ultrasensitive and spatial proteomics

In this work, Davis et al. described a spatially resolved proteome profiling effort on a tissue slice of a human brain tumor by combining laser capture microdissection and LC-MS/MS. The application of spatial proteomics on human brain tumor tissue section to phenotyping cellular heterogeneity at the proteome level is interesting. Nonetheless, I have several major concerns of this work.

1. The overall work does not offer much innovation or analytical advances in spatial proteomics in the aspects of spatial resolution or proteome coverage. Spatial proteome mapping of mouse brain tissue at 1000 um resolution was published in Genome Research, 2007 (doi:

10.1101/gr.5799207). Recent work in Nature Communications

(<https://www.nature.com/articles/s41467-022-35367-2>) demonstrated a beautiful deep proteome mapping of mouse brain. The current work does not provide substantial advances over early work.

2. the biological insights appear to be also limited, despite the interesting application to the tumor tissue. The main reason perhaps is related to the relative poor spatial resolution (833um and 350 um).

3. The data quality and robustness of the workflow are also of concern. As in Figure S2B, Why there are so few proteins quantified in all pixels. Some pixel quantified very few proteins, which could be an analytical failure. It is puzzling to me why there are so much missing data for the majority of the proteins. What are the QA/QC measures used to ensure the data quality?

Minor concerns:

a. The correlation between IHC image and proteome maps are great. But it will be better to show a large area of the IHC images to show the correlation in a bigger field of view (as supplemental)

b. the UMAP plot in Figure 4 does not show clear separations of clusters. It is unclear how the clusters were defined.

c. For data analysis, the Match Between Runs Algorithm is known to greatly increase proteome coverage. It will be good to show show the # of IDs without match between runs in Fig S2.

d. Detailed proteome identification and quantification results obtained from high-resolution data (96 pixels of 350 μm) was not presented.

e. In Fig 5 and p7 line 18, authors demonstrated that "Clustering in high- and low-resolution maps was broadly consistent (Figure 5A) with 19 generally contiguous clusters that represent the solid tumour (cluster 1), the brain/tumour 20 interface (cluster 3), the margin between (clusters 2 & 5), and blood vessels (cluster 6)." But it is hard to see how's the data consistent between high and low resolution data. Additionally, it would be helpful if the authors could provide a UMAP visualization of the high-resolution data.

POINT-BY-POINT REPLY

In the following response, Reviewer's comments are presented in black text and our responses are in red text. As we discussed below, we have added new data to the manuscript and removed the data relating to MALDI imaging of lipids. The mapping between the original figure numbers and their new locations is shown in the below table for clarity.

New Figure Number	Original Figure Number
Figure 1	Figure 1
Figure 2	Figure 2
Figure 3	Figure 4
Figure 4	Figure 5
Figure 5	N/A
Figure 6	Figure 6
Figure S1	N/A
Figure S2	Figure S1
Figure S3	Figure S2
Figure S4	Figure S3
Figure S5	Figure 3
Figure S6	Figure S4
Figure S7	Figure S5
Figure S8	N/A
Figure S9	Figure S7
Figure S10	Figure S8

Reviewer #2 (Remarks to the Author): Expert in AT/RT genomics, clinical research, and tumour microenvironment

“Davis et al. report on a workflow for spatially-resolved, quantitative proteomics. As I am not an expert on the technical parameters of spatial proteomics, I comment on the ATRT part. If this manuscript aims to offer new insights on ATRT, I have some questions/comments, which needs to be addressed.”

We would like to thank this reviewer for their assessment of our manuscript with view on ATRT pathology. We would like to point out the new structure of the manuscript, where the original Figure 3 containing immunohistochemistry validation has been moved to supplementary, the original figures 4 and 5 are now marked figures 3 and 4. We have added new data that is now marked figure 5, showing data collected at 40 µm resolution.

We have added text on lines 61-65 clarifying the aim of this study in the context of ATRT Biology, which reads:

“The principal aim of the study was to design a workflow for deep proteomics and novel spatially-aware statistics for tissue feature discovery, rather than provide an in depth analysis of AT/RT biology, which will be the subject of a follow-up ‘multiomics’ study on paediatric brain tumours. The AT/RT was chosen based on tissue quality, homogeneity and abundance.”

1. *“The genetic and histologic hallmark of ATRT is the loss of SMARCB1. Please provide a SMARCB1 staining. How does this protein behave at tumor versus non-tumor regions?”*

The reviewer raises an important point. The proteome depth detected in our data was not sufficient to pick up presence of SMARCB1 within the tumour or its periphery, implying low abundance/absence of SMARCB1 in the analysed area. We made an attempt to validate the spatial context of SMARCB1 loss in the tissue by IHC using an array of commercially available antibodies. Unfortunately, none of the antibodies were able to bind SMARCB1 in the used fresh frozen tissue. These antibodies had been used on FFPE tissue successfully before, which is why we assume that they exclusively bind the denatured form of SMARCB1. The case showed loss of SMARCB1 staining at diagnosis. We have now included a new Figure S1, which shows clinicopathological data including SMARCB1 IHC of a preoperative tumour biopsy of the patient.

2. *“ATRT are subdivided into three molecular distinct groups (ATRT-TYR, ATRT-SHH, ATRT-MYC). Into which subgroup falls this tumor (by methylation arrays).”*

The tumour was subtyped as ATRT-SHH by methylation profiling during the diagnostic investigation. We have now included this information in the manuscript (line 89-91) which reads:

“The tumour was fully characterised in vivo, corresponding to a supratentorial AT/RT with nuclear loss of SMARCB1 protein and corresponding to methylation class AT/RT-SHH (Figure S1)”

3. *“ATRT are known to have some heterogeneity on histopathological level. It would be interesting to see how the heterogeneity in protein expression is related to this histological heterogeneity.”*

This is a very important point raised by the reviewer. The analysed specimen that was selected for our study was deliberately chosen to be relatively homogenous in classical tissue histology terms as it did not show much heterogeneity based on microscopy H&E staining. The rationale is rooted in our belief that spatial proteomics will be able to discern spatial and systematic heterogeneity at proteome level even in relatively homogenous tissue samples and in correlation with observable and hidden features (by H&E stain). To this end, our spatial proteomics map including advanced bioinformatics workflows revealed distinguishable sub-sections with differential proteome profiles (see also comments below) in correlation with ECM of the tumour or even proximity to blood vessels and therefore sources of nutrients (new added data, Fig 5).

In addition to the supplied H&E stains across different sections of the tissue, we also now added another stained section in Figure S1, showing relative homogeneity of the specimen.

4. *“The authors report on differences in CD45 protein expression within the sample. Also this result could be easily confirmed by a staining.”*

We thank the reviewer for highlighting this point. We agree that this would add value to our work, and we did include CD45 staining in the original manuscript (Figure 3, panel F). However we have moved this figure to the supplementary material as figure S4F to allow for the inclusion of the new figure 5 data showing data collected at 40 μm resolution. The abundance profile of CD45 staining is in very good agreement with the detection of CD45 by mass spectrometry.

5. *“General concern: the regions which are analyzed with this technic are huge. Multiple cells (hundreds of cells) might be detected within one region. Might the authors comment on what can be the advantage of this platform in comparison to many other platforms on the market, which offer spatial protein analyses with a higher resolution.”*

We agree with the reviewer that smaller regions (higher resolution) would be preferable and is available at subcellular on multiplexed immunohistochemistry platforms with either imaging or mass cytometry-based detection. However, these platforms are only capable of multiplexed detection of 10's of proteins, while mass spectrometry-based proteomics can detect 1000's of proteins per sample. The regions analysed comprise about 4000 cells as shown in our titration experiment (supp. Figure S2) in the first and lowest resolution (833 μm per spatial resolution), ~800 cells in the medium resolution (350 μm spatial resolution) and only ~5-10 cells in the newly added data (Figure 5, 40 μm spatial resolution). However, we need to consider proteome depth and also analytical burden. Smaller voxels would result in lower protein numbers. It is a matter of debate; how many proteins need to be identified to make meaningful functional deductions from the data and not just to differentiate cell types. We agree with other experts in the field that ~4000 proteins depth would allow us to make meaningful conclusions at a functional level and not just differentiate regions

within the tumour. On the other hand, higher resolution would increase the sample number quadratically. Already there are not many proteomics studies described in the literature to date that include more than 400 samples and no spatial proteomics studies that come close to this number. Going forward, spatial resolution of this technique will improve as more sensitive equipment and high throughput methods become available. The presented data here is state-of-the-art.

In the newly added data at 40 μm resolution the tissue voxels comprise sections of approximately 5-10 cells (Figure 5). At this resolution, we detect approximately 1.5k proteins, a considerable improvement in multiplexed protein detection in comparison to antibody-based approaches.

Spatial transcriptomics platforms offer higher resolution (single cell) but are biased by way of detecting panels of transcripts of interest. Although not at single cell resolution, our data surpasses what is currently provided by spatial transcriptomics platforms such as vizium. In the proteomics space there are affinity reagent based methods, which offer higher spatial resolution, but again only on a small subset of selected proteins. . However, these methods will not be able to be used in an unbiased discovery type approach and are largely untested/unchallenged with regards to false positive rates and proteoform distinction.

Reviewer #3 (Remarks to the Author): Expert in brain cancer genomics, tumour microenvironment, and laser capture microdissection

“The present study is original and well-conducted. The authors present a workflow for spatially-resolved, quantitative proteomics of tissue that generates maps of protein abundance across tumor tissue sections and employs spatially-aware algorithms that do not require prior knowledge of the fine tissue structure to detect proteins and pathways with spatial abundance patterns. I am a strong believer in such approaches that tried to diverge from what we are used to seeing and try to identify new information based on the intrinsic molecular characteristics of the tissue.”

We would like to thank this reviewer for taking their time to thoroughly assess our manuscript. We are glad that the reviewer agrees with our systematic approach and values our approach for unbiased detection of spatial protein expression patterns. We hope that the newly added data shown now in figure 5, specifically in dependence of proximity to nutrient/oxygen sources (blood vessels) will be of great interest to this reviewer.

“ I believe strongly that many of the answers we are seeking are in the tissues section, which represents snapshots of the tumor behavior in vivo. This work represents an attempt to look for these answers utilizing an approach that is innovative in its combination of well know approaches.”

We appreciate that the reviewer recognizes the novelty in our combined approach, specifically in the context of applying it to the brain cancer field. We agree that this approach has huge potential for the molecular level analysis of tumours

“The paper needs some modifications. It seems generally that there was a lack of care in presenting the figures that interfered with clarity.”

1) *“rephrase sentences 34,35,36,37. It is not clear.”*

We have rephrased the sentences to improve clarity. We have described the MASP-Proteomics study by Ma et al and reworded the above lines. The modified text now reads:

“Additionally, Ma et al. used a micro-scaffold to cut 1 mm-thick sections of mouse brain at 400 μm resolution prior to quantifying 5,000 proteins using LC-MS/MS. However, these approaches have primarily emphasized visualizing protein abundance, neglecting the potential of utilising spatial relationships between samples to uncover novel spatial features, an approach frequently employed in spatial transcriptomics methodologies.” (Lines 59-64)

2) *“Figure 1 needs more details to be understood, needs a,b,c letters with explanations in the legend.”*

We have added some explanation to the legend and adjusted the figure composition to improve clarity. The figure legend now reads:

“Tissue is mounted onto a slide compatible with laser capture microdissection (LCM). A general overview is depicted in panel (A): Tissue is segmented into a regular grid shape (Annotate), and each element of the grid is isolated by LCM into a well of a 96-well plate (Cut). Proteins from each sample are lysed in RIPA buffer (Lyse) and digested into peptides (Digest) before analysis by LC-MS/MS. The quantitative information for each protein can be mapped back to its location within the gridded tissue and visualised in a topographic protein map, with one map per protein quantified (Proteomic Maps). This workflow was applied to an Atypical Teratoid-Rhabdoid Tumour (AT/RT). (B) A H&E-stained section was segmented into a 24x16 grid and analysed with the workflow to generate a Protein Topography Stack containing over 5,000 proteomic maps at 833 μm resolution, allowing for the resolution of several features within the tissue while maintaining good throughput. (C) We then proceeded to apply this workflow at smaller length scales. In total, we applied this workflow over three length scales within serial sections of the AT/RT tumour tissue: 833 μm resolution, covering an entire tumour section; 350 μm resolution, covering part of the boundary between two visibly distinct regions; and 40 μm resolution, covering several blood vessels and their surrounding cells. These data relating to these three length scales are shown in Figures 2, 3, 4 and 5, respectively. Scale bar represents 833 μm in the bottom, 350 μm in the middle image, and 40 μm in the top image.”

3) *“Figure S2 is not explained clearly”*

We have clarified the information conveyed in Figure S2, now Figure S3, the main text referring to this now reads:

“In total, 5,321 proteins were identified, with 32 – 4,741 proteins identified per sample (Figure S2A). This range of proteins identified per sample includes empty pixels where no tissue was located within a pixel, demonstrating a low level of contamination throughout the workflow. These empty pixels are why only a few proteins were quantified in all 384 pixels in Figure S2B.”

The figure legend of now reads:

“(A) 833 μm resolution data. Violin plots showing distributions of the number of identified and quantified proteins per voxel. Solid vertical lines represent the median value. Dashed vertical lines

represent upper and lower quartiles. (B) 833 μm resolution data. Histogram showing the number of voxels where each protein was identified/quantified as per panel (A). (C) 350 μm data as in panel (A). (D) 350 μm data as in panel (B). (E) 40 resolution μm data. Violin plot shows distribution of the number of quantified proteins per voxel. Solid vertical line represents the median value. Dashed vertical lines represent upper and lower quartiles. (F) 40 μm resolution data. Histogram showing the number of voxels where each protein was quantified.”

- 4) *“Sentences from 65 to 70: why did you choose these genes PYGL, HBB PRPH, and H4? Put them in the same order as in the figure. Please write the abbreviation of the genes also in the figure”*

The selected genes/proteins were chosen as examples for proteins showing positive autocorrelation and up-regulation in tumour (PYGL, also putative cancer marker in colorectal cancer) or periphery (PRPH, also a classical neuronal marker), no autocorrelation (H4, indicating similar DNA abundance/cell numbers) and focussed/feature dependent presence (HBB collocating with blood vessel). We agree that this is not immediately apparent so have added the rationale to the text (lines 103-108) and modified the figure as requested.

This now reads: “Figure 2A shows proteomic maps for four example proteins, haemoglobin (HBB), histone H4 (HIST1H4A), peripherin (PRPH) and liver glycogen phosphorylase (PYGL). The selected proteins were chosen as examples showing: positive autocorrelation and increased abundance in tumour (PYGL) or periphery (PRPH); no autocorrelation (HIST1H4A, indicating similar DNA abundance/cell numbers); and a protein which should correlate with the large region of haemorrhage visible in the upper-left region of the section (HBB).”

- 5) *“Figure S3 seems not the right one according to the text.”*

The reference to figure S3 is in the intended place but we agree that the explanation needs clarification. In the revised manuscript, this is now figure S4.

The text referring to figure S4 (lines 83-88) now reads:

“To determine whether the observed spatial variability could have been caused by systematic impacts derived from tissue sampling and processing, the summed and mean intensity per voxel were inspected and are shown to be visibly consistent across the tissue section (Figure S4A,B). This demonstrates that no sampling bias was introduced and protein abundance differences are mostly compositional.”

- 6) *“Figure 3 needs better referring in the text and authors should point out that the histological images are close-ups of the region of the proteomic map”*

We have now added a paragraph and some background to the text in order to highlight the figure and its relevance for validation further. We also explain the fact that these are close-ups (lines 152-161). It now reads:

“Three proteins showing significant spatial variation were selected for follow-up immunohistochemistry (IHC) staining and are presented side-by-side with their proteomic maps: glycogen phosphorylase (Figure S5A, B), aspartate beta-hydroxylase (ASPH) (Figure S5C, D) and CD45 (PTPRC) (Figure S5E, F) to validate the spatially resolved protein expression data generated above. The location within the proteomic maps of the presented IHC images are marked by the black boxes”

7) *“Figure 4a needs to be subdivided into sub-panels for better clarity. Authors take things too much for granted and the reader could get lost if the figures are not well explained. Draw some arrows to highlight boundaries”*

We have now tried to reduce the complexity in this figure and have added some more background. We have reformatted the figure to show the H&E stain and two rows of panels. The two rows of panels both show clustering, a UMAP plot and geneset enrichment per cluster. The first row's clusters are defined in a spatially unaware method (hierarchical clustering). The second row's clusters are defined in a spatially-aware method (affinity network fusion). The figure legend has been updated to reflect these changes. The text referring to this figure was significantly modified to reflect these changes (lines 152 - 174 & 187 - 202).

8) *“Figure S4 needs H&E images for better clarity”*

We have added an H&E image of the tissue to help orient the reader within this figure.

9) *“The Maldi Imaging section again is not well explained through Figure S6, which is not clear and it seems not to add more information”*

We thank this reviewer for raising this point. Including the MALDI approach was an attempt to showcase further improvements to our approach further down the line in order to include spatial lipidomics at a higher spatial resolution and to illustrate its power for ML approaches going forward. However, we appreciate that this is somewhat separated from the main paper. Therefore we have taken this aspect out of the paper in order to improve clarity and remove a distraction from the main story.

“The approach presented in the paper seems sure laborious, however, if any research group can have access to the technology proposed, it has a strong potential for identifying new leads in cancer worth it to pursue. It seems very demanding to apply this approach to many tissue sections across cohorts of patients-derived tissues. Authors should highlight this limitation and how they would envision its application for identifying robust targets. It is more of a technical paper since it does not leave so far useful scientific information that could be used considering that it is done on one single tissue section.”

We thank the reviewer for this positive assessment. We completely agree that this technique has the potential to be widely used in the area of digital pathology. We also agree that in its current form it is mostly a discovery tool and not so much applicable for diagnosis. However, we believe that based on the atlas data we provide here and using ML approaches going forward, easier accessible data (IHC, transcriptomics, etc.) can in the future be used to impute protein level spatial abundance data if cell type distribution can be mapped. Together with other groups working in this new field and generating single cell/phenotype data, we anticipate that characterisation of a few disease specific markers in combination with atlas data will be able to derive deep proteomes within pathogenic tissue with much reduced analytical effort.

Reviewer #4 (Remarks to the Author): Expert in ultrasensitive and spatial proteomics

“In this work, Davis et al. described a spatially resolved proteome profiling effort on a tissue slice of a human brain tumor by combining laser capture microdissection and LC-MS/MS. The application of spatial proteomics on human brain tumor tissue section to phenotyping cellular heterogeneity at the proteome level is interesting. Nonetheless, I have several major concerns of this work.”

We thank this reviewer for taking the time to assess our manuscript. It is encouraging that they see this as interesting approach, implying that it is novel and genuine, and in agreement with reviewer #3.

1. *“The overall work does not offer much innovation or analytical advances in spatial proteomics in the aspects of spatial resolution or proteome coverage.”*

Our approach is the first description of an unbiased spatial and systematic approach for spatial proteomics in a disease context and on clinically relevant material. Previous approaches used biased (feature selected) areas, introducing confirmation bias and/or did not make use of the spatial information contained in the data beyond cartography. We also show the first application of spatial autocorrelation analysis and spatially-aware clustering approaches on proteomics data and reveal immune system engagement plus ECM alterations in a rare paediatric tumour which are clearly avenues into treatments for these otherwise cold/untreatable rare tumours in young children. In the updated manuscript we now include a further increase of spatial resolution, which again allows a novel type of analysis, not undertaken before, namely, the correlative dependencies of spatial protein expression in phenotypically identical cancer cells in context of proximity to nutrients and oxygen (blood vessels). However, the resolution will continue to be limited/ dictated by the desired proteome depth (see also titration experiment in figure S2) and sample throughput /analytical burden. In summary we present data and approaches to spatial analysis of proteomics data which have not yet been reported and offer the potential to direct translation into the clinic, rather than based on a pure academic approach on model organisms and easily accessed tissue with obvious and known heterogeneity.

Spatial resolution and proteome coverage are also addressed in the subsequent points.

Spatial proteome mapping of mouse brain tissue at 1000 um resolution was published in Genome Research, 2007 (doi:

10.1101/gr.5799207).

This referenced work describes the analysis of about 70 voxels of mouse brain with a side length of 1 mm³. While this is a heroic effort, which pioneers the technique of using LCM with LCMS, the tissue volume analysed in our work per voxel is 144x and 816x smaller. Our newly added data at 40 µm x 40 µm x 10 µm spatial resolution reflects an improvement of 62,500x in analysed volume per voxel. The referenced work is descriptive and the authors did not undertake autocorrelation analysis or spatially aware and functional clustering. Consequently, they cannot derive new functional data

from this descriptive feasibility study. This citation makes a good argument for the novelty of our approach, which is why we have now included it in our manuscript (reference 34, lines 46-48).

In our manuscript we advance this pioneering work and subsequent published work significantly and provide an analytical framework which directly allows us to derive functional data in an unbiased approach. We introduce completely new concepts into the area of spatial proteomics, which are desperately needed to advance this novel and niche area of proteome research and facilitate translation to clinically and biologically relevant questions in health and disease.

Lines 46-48 read: “For instance, Petyuk et al. sampled seventy, 1 mm cubes of mouse brain to generate a spatial proteome at a depth of approximately 1,000 proteins”

Recent work in Nature Communications (<https://www.nature.com/articles/s41467-022-35367-2>) demonstrated a beautiful deep proteome mapping of mouse brain. The current work does not provide substantial advances over early work.

We cannot but disagree with the reviewer about this comment. While the referenced paper certainly provides some information into the proteomic landscape of the mouse brain, the technical and analytical approach is very limited. Looking into the details of the cited work, we found that the voxel side length is similar, the thickness of the used tissue is a factor 100 (!) larger, in a specimen that is by a factor of 1600 smaller in scale to the human brain. In combination the referenced study has a 160000 fold lower resolving power. However, we apply spatial proteomics to clinically relevant, human cancerous brain tissue and go way beyond mere protein topography. In the referenced study, no spatial analysis is undertaken beyond reporting/quantitation of proteins across the analysed voxels. No attempt is made to map detected proteins to structural and functional features in the mouse brain. The analysis does not go beyond matching proteins to pathways.

Even from the technology aspect, the referenced study is very different as they report data from 208 voxels, each requiring at least three hours of mass spectrometry acquisition time, resulting in at least 26 days from start to finish. Our approach reaches the same proteome depth, from 100x less input material and at a 4.5x-higher throughput (38 minutes per sample). While the work makes use of spatial analysis of individual pathway members, no attempt to analyse autocorrelation or spatially aware clustering is made. The authors merely map proteins to known regions in the mouse brain (confirmation bias). There is no connection to human disease let alone usage of a mouse disease model, or on how spatial proteomics could be used to analyse disease. This is a missed opportunity on their behalf. Technically, cross contamination across voxels is a major concern using their mechanical device, which in any case is not easy to reproduce compared to the relative accessibility of laser capture microscopes which are produced by at least four manufacturers and are widely available in pathology labs. In summary, our approach could not be more different from the one referenced. This covers all aspects one could consider including sampling, data generation and analysis.

We have reworked the part of our introduction which refers to previous work in the field. Lines 45-56 now read:

“However, sampling in a systematic manner, like MSI, could reveal novel tissue fine-structure and give insights into spatial protein expression patterns. For instance, Petyuk et al. sampled seventy, 1 mm cubes of mouse brain to generate a spatial proteome at a depth of approximately 1,000

proteins³⁴. Piehowski et al. used LCM-proteomics to sample a feature-rich landscape of mouse uterine tissue in a rastered grid at a resolution of 100 μm and used their custom, robotic, nanolitre-scale nanoPOTS sample preparation platform to quantify over 2,000 proteins across 24 voxels³⁵. Additionally, Ma et al. used a micro-scaffold to cut 1 mm-thick sections of mouse brain at 400 μm resolution prior to quantifying 5,000 proteins using LC-MS/MS²⁷. However, these approaches have primarily emphasised visualising protein abundance, neglecting the potential of utilising spatial relationships between areas of correlated protein expression and have been unable to discover novel spatial features, an approach frequently employed in spatial transcriptomics methodologies.”

2. the biological insights appear to be also limited, despite the interesting application to the tumor tissue. The main reason perhaps is related to the relative poor spatial resolution (833um and 350 um).

We present a very large dataset with sufficient spatial resolution to discover new features in a human brain tumour. The selected resolutions mark “sweet spots”, yielding sufficient depth for feature/functional discovery and at the same time a manageable analytical burden with regards to sample numbers. Not long ago, even this sample number would have been unmanageable. Our data clearly show that the resolution is sufficient to discover unknown areas within tumour tissue and inform about new biomarkers, drug targets and avenues for treating these fatal tumours.

We agree that higher spatial resolution would be an advantage. In order to address this point and further introduce yet another novel strategy, we increased spatial resolution in newly added data and added a new spatial analysis approach correlating the spatial proteome with proximity to blood vessels as sources of nutrients and oxygen, which is highly relevant in tumour context. The spatial resolutions reported in our manuscript are orders of magnitude higher than the papers referenced by the reviewer. We hope that the newly added data using a spatial resolution of 40 μm addresses this reviewer's point.

This data is presented in what is now figure 5. The corresponding text on lines 214 - 247 reads:

“High resolution proteomic maps visualise proteomic patterns around blood vessels

We further focussed on an area containing four blood vessels, represented in a single voxel in the previous analysis (350 μm spatial resolution), allowing us to map protein abundance patterns to potential nutrient/oxygen gradients within the tumour tissue by proxy of distance of individual cells to blood vessels.

This region was subdivided into a 9-by-9 grid, resulting in 40 μm spatial resolution (Figure 5A). Each 40 μm x 40 μm x 10 μm voxel contained between 5 - 10 visible nuclei. In total, 1,550 proteins were quantified using data-independent analysis (DIA-PASEF⁴⁵) on a Bruker timsTOF SCP at a throughput of 40 samples per day.

We then measured the distance of each cell to the nearest blood vessel (Figure 5B) to use this distance as a proxy for nutrient and oxygen availability. As expected, we observed positive correlation of blood proteins with closeness to blood vessels (haemoglobin, Figure 5C)). After spatial clustering as above, the tissue is segmented into four main clusters (Figure 5D), representing voxels further from blood vessels (cluster 1), and voxels immediately next to or containing blood vessels (clusters 2, 3 & 4). UMAP dimensionality reduction shows general separation between voxels within

cluster 1 and other voxels (Figure S8A,B,C). Testing for enrichment of MSigDB hallmark gene sets within the clusters shows an enrichment of the “Oxidative Phosphorylation” term within cluster 1, and terms related to blood within clusters 2 & 4 (Figure S8D).

GAPDH shows consistent intensity across the tissue (Figure 5E). Two main patterns are visible in the generated proteomic maps in Figure 5E: proteins correlated to voxel:blood vessel distance and proteins anticorrelated to voxel:blood vessel distance. Alpha-2-Macroglobulin, a highly abundant blood protein forms an intensity gradient around blood vessels, correlating with haemoglobin. Aspartate beta-hydroxylase (ASPH) shows highest intensity in those voxels furthest away from blood vessels, consistent with observations that ASPH can be regulated by hypoxia^{46,47}. Peroxiredoxin 3, suggested to protect cells against apoptosis caused by oxidative stress in hypoxia⁴⁸, Protein disulfide isomerase family A members 4 and 6, suggested to contribute to avoidance of cell death pathways in some tumours and activation of Wnt signalling^{49,50}, also show increased intensity in voxels further away from vessels, indicating the presence of a cancer proteo-phenotype within the tumour tissue and in dependency of oxygen/nutrient availability in otherwise homogeneous tissue. In addition, three erythrocyte markers show specific, high intensity within voxels containing, or that are very close to blood vessels: Solute carrier family 4 member 1, Spectrin Beta, and Ankyrin 1⁵¹⁻⁵³.

”

3. The data quality and robustness of the workflow are also of concern. As in Figure S2B, Why there are so few proteins quantified in all pixels. Some pixel quantified very few proteins, which could be an analytical failure. It is puzzling to me why there are so much missing data for the majority of the proteins. What are the QA/QC measures used to ensure the data quality?

We thank the reviewer for this comment and have added more detail to what is now Figure S3. We have split the proteins quantified and identified into those proteins identified only by MS/MS spectra, identified by MS/MS spectra or Match Between Runs (MBR) and those proteins with quantification values in MaxQuant’s “Intensity” column, and those with values in the “LFQ Intensity” columns. These additional data clearly show in Figure S3B that the reason for missingness is the high stringency of quantification performed by the MaxLFQ algorithm. This approach is very conservative and avoids artificially increasing protein numbers and data completeness by missing number imputation, which unfortunately is observed more frequently in the field. In our view, the observations made by the reviewer in our manuscript, highlights the stringency we applied to present robust and not inflated numbers.

Additionally, we would like to draw the reviewer’s attention to Figure S3 A&B of the original manuscript (now figure S4 A&B). There we show that the aggregate intensity is very consistent across the pixels measured that do not cover the region of haemorrhage or pixels where no tissue was present. Furthermore, we chose not to perform any type of imputation on this dataset. The handling of missing values within proteomics data is still an active area of research and debate, and there are no studies available that investigate the combination of missing values, their imputation in combination with the spatial locations of said missing values (<https://www.nature.com/articles/s41598-021-81279-4>, <https://pubs.acs.org/doi/10.1021/acs.jproteome.5b00981>, <https://www.nature.com/articles/s41598-022-04938-0>).

Admittedly, some voxels identify lower numbers. However, these voxels predominantly contain clotted blood or are at the grid edge/corner and contain no tissue. We could have excluded these voxels to escape such criticism, but decided to be transparent about and conservative in our approach. Our results are exactly as expected/desired, an empty voxel should contain minimal protein amounts, therefore demonstrating a low level of system carry-over and contamination.

Minor concerns:

a. The correlation between IHC image and proteome maps are great. But it will be better to show a large area of the IHC images to show the correlation in a bigger field of view (as supplemental)

We have now added a larger field of view for the IHC stainings to visualise the abundance of selected proteins across the whole tissue section. This data can be found in supplemental data 4.

b. the UMAP plot in Figure 4 does not show clear separations of clusters. It is unclear how the clusters were defined.

No clustering was performed on the UMAP-embedded data, the colours within this plot correspond to the sample colours within the spatial cluster map shown in figure 3B. The intention of this figure was to provide an alternate visualisation of sample clustering and that these relationships are generally conserved after dimensionality reduction.

c. For data analysis, the Match Between Runs Algorithm is known to greatly increase proteome coverage. It will be good to show show the # of IDs without match between runs in Fig S2.

As mentioned above, this is now included in Figure S3. To reiterate this important point, by using Max LFQ values, we report the minimal number of proteins in each pixel. This makes sure that no noise is overrepresented as protein intensities. Therefore missing values can be assumed to be resulting from spatial expression differences as expected in real tissue.

d. Detailed proteome identification and quantification results obtained from high-resolution data (96 pixels of 350 μm) was not presented.

We have now included this data in the manuscript submission files in addition to the already deposited tables as part of our PRIDE data disposition. They are supplemental data files 1,2 and 3 - relating to the 833, 350 and 40 μm data respectively.

e. In Fig 5 and p7 line 18, authors demonstrated that "Clustering in high- and low-resolution maps was broadly consistent (Figure 5A) with 19 generally contiguous clusters that represent the solid tumour (cluster 1), the brain/tumour 20 interface (cluster 3), the margin between (clusters 2 & 5), and blood vessels (cluster 6)." But it is hard to see how's the data consistent between high and low resolution data. Additionally, it would be helpful if the authors could provide a UMAP visualization of the high-resolution data.

We have now added a UMAP visualisation at every analysed spatial resolution. The observed consistency is related to the general ability to distinguish the mentioned tumour interface areas based on protein clustering. We have now reordered the figures to make this conclusion more intuitive.

REVIEWERS' COMMENTS

Reviewer #2 (Remarks to the Author):

The authors addressed the questions/comments I raised before. Eventhough the technic have a clear limitation in the resolution, I am convinced that it provides some novelty. I have no further questions. I recommend this paper for publication.

Reviewer #3 (Remarks to the Author):

Upon a thorough examination of the rebuttal letter and a meticulous review of the responses provided by the authors, I am pleased to convey my growing confidence in the way in which the authors have adeptly and comprehensively addressed the concerns I initially raised. It is evident that they have taken my feedback seriously and made substantial efforts to refine their work, resulting in a significantly improved manuscript reinforcing its potential to make a valuable contribution to the field.

Furthermore, I would like to suggest an additional enhancement to the manuscript. In light of the article's thematic relevance, it might be advantageous for the authors to include a reference to another research paper recently published in this very journal (Zheng et al (2023) 14:4122). This particular article, which surfaced in June 2023, adopts a similar investigative approach but leans on transcriptome data analysis. By mentioning and briefly discussing this complementary study, the authors can effectively underscore the broader implications and synergies within the field, thus enriching the context of their own work.

In conclusion, I recommend the acceptance of this manuscript for publication in your esteemed journal

Reviewer #4 (Remarks to the Author):

I appreciate the authors' substantial efforts in improving this manuscript in this resubmission. It is interesting to see the demonstration of spatial proteome mapping on the tumor microenvironment at the three different resolution to cover the entire tissue section at 833 um resolution, 350 um resolution covering a boundary of distinct regions, and 40 um resolution covering several blood vessels. The demonstration of spatial proteomics in a disease context with multiple spatial resolution is the novel aspect of this work.

Minor concerns:

1. The main limitations of this workflow include 1) sensitivity and missing data, and 2) the lack of throughput to cover large mapping area if higher resolution is pursued (e.g., 40 μm). Maybe the authors should consider a section discussing the limitations of current workflow and future perspectives.
2. Figure 2, it is not straightforward to see the relationship between A and B. If B is part of A, could the protein maps of B be highlighted as which part in the maps in A?
3. The three erythrocyte markers should be highlighted in Fig.5

POINT-BY-POINT REPLY

In the following response, Reviewer's comments are presented in black text and our responses are in red text.

Reviewer #2 (Remarks to the Author):

The authors addressed the questions/comments I raised before. Eventhough the technic have a clear limitation in the resolution, I am convinced that it provides some novelty. I have no further questions. I recommend this paper for publication.

We would like to thank this reviewer for the positive assessment and useful comments through the process.

Reviewer #3 (Remarks to the Author):

Upon a thorough examination of the rebuttal letter and a meticulous review of the responses provided by the authors, I am pleased to convey my growing confidence in the way in which the authors have adeptly and comprehensively addressed the concerns I initially raised. It is evident that they have taken my feedback seriously and made substantial efforts to refine their work, resulting in a significantly improved manuscript reinforcing its potential to make a valuable contribution to the field.

Furthermore, I would like to suggest an additional enhancement to the manuscript. In light of the article's thematic relevance, it might be advantageous for the authors to include a reference to another research paper recently published in this very journal (Zheng et al (2023) 14:4122). This particular article, which surfaced in June 2023, adopts a similar investigative approach but leans on transcriptome data analysis. By mentioning and briefly discussing this complementary study, the authors can effectively underscore the broader implications and synergies within the field, thus enriching the context of their own work.

In conclusion, I recommend the acceptance of this manuscript for publication in your esteemed journal

We would like to thank this reviewer for the positive assessment and useful comments through the process. We agree that the mentioned study by Zheng et al. justifies special mention in the context of our manuscript as it provides a roadmap for follow up work in the proteomics space, incorporating machine learning approaches and other OMICS modalities in integrative approaches. We have now added the following paragraph to the discussion section of our manuscript beginning on line 289:

“Spatial transcriptomics studies such as Zheng et al have started using machine learning to derive diagnostic and prognostic value in several oncology contexts, indicating that spatial ‘omics data can aid pathological assessment of tumour tissue at the point of care. Integration of this approach with other ‘omics modalities such as proteomics, lipidomics and metabolomics has the potential to further refine this approach, deriving mechanistic information about an individual’s pathology.”

Reviewer #4 (Remarks to the Author):

I appreciate the authors' substantial efforts in improving this manuscript in this resubmission. It is interesting to see the demonstration of spatial proteome mapping on the tumor microenvironment at the three different resolution to cover the entire tissue section at 833 um resolution, 350 um resolution covering a boundary of distinct regions, and 40 um resolution covering several blood vessels. The demonstration of spatial proteomics in a disease context with multiple spatial resolution is the novel aspect of this work.

We would like to thank this reviewer for the positive assessment and useful comments through the process.

Minor concerns:

1. The main limitations of this workflow include 1) sensitivity and missing data, and 2) the lack of throughput to cover large mapping area if higher resolution is pursued (e.g., 40 um). Maybe the authors should consider a section discussing the limitations of current workflow and future perspectives.

We had included future perspectives, specifically with view of machine learning approaches and limitations (throughput/resolution) within the final two discussion paragraphs. To address the reviewer's comment, we have now expanded on these limitations (In combination with the above revision based on Reviewer 3's comment), beginning on line 365 of the revised manuscript:

"A limitation of our study is that there is the requirement to strike a balance between the need for spatial resolution and the need for meaningful depth to cover pathways of interest, these limiting factors are driven by technical limitations such as sensitivity and throughput of the LC-MS systems used. Increasing the spatial resolution demonstrated here towards the single-cell level and covering comparable areas poses a formidable analytical challenge, which further escalates when analysing tissue in three dimensions. Several recent studies have developed methods towards increased sensitivity and throughput of low-abundance samples, Kreimer et al demonstrate an efficient dual trap-column liquid chromatography configuration for the analysis of single cells, and Derks et al and Thielert et al have both demonstrated 3-plex multiplexed methods of data independent analysis, with both frameworks offering the potential for future increases in multiplexing. The similar analytical demands of systematic spatial proteomics analysis and single cell proteomics will allow both fields to develop in lock-step. Furthermore, the current trend in commercial mass spectrometry for proteomics towards faster, more sensitive instruments will also likely provide benefits."

2. Figure 2, it is not straightforward to see the relationship between A and B. If B is part of A, could the protein maps of B be highlighted as which part in the maps in A?

We agree that this connection of B being a subsection of A is an important point and had indeed highlighted this relation within the top panel of figure 2. However, to improve clarity be added a simple annotation of the 2 main regions to B (“Solid tumour, tumour periphery, blood vessel”) and added a stylized image to further explain the relation schematically

3. The three erythrocyte markers should be highlighted in Fig.5

We have now added a * to erythrocyte markers and defined this in the figure legend.